# Dynamics of venom composition across a complex life cycle

**Yaara Y Columbus-Shenkar[1†], Maria Y Sachkova[1†], Jason Macrander[2], Arie Fridrich[1], Vengamanaidu Modepalli[1], Adam M Reitzel[2], Kartik Sunagar[1,3], Yehu Moran[1]***

[1]Department of Ecology, Evolution and Behavior, Alexander Silberman Institute of Life Sciences, The Hebrew University of Jerusalem, Jerusalem, Israel; [2]Department of Biological Sciences, University of North Carolina at Charlotte, Charlotte, United States; [3]Evolutionary Venomics Lab, Centre for Ecological Sciences, Indian Institute of Science, Bangalore, India

**Abstract** Little is known about venom in young developmental stages of animals. The appearance of toxins and stinging cells during early embryonic stages in the sea anemone *Nematostella vectensis* suggests that venom is already expressed in eggs and larvae of this species. Here, we harness transcriptomic, biochemical and transgenic tools to study venom production dynamics in *Nematostella*. We find that venom composition and arsenal of toxin-producing cells change dramatically between developmental stages of this species. These findings can be explained by the vastly different interspecific interactions of each life stage, as individuals develop from a miniature non-feeding mobile planula to a larger sessile polyp that predates on other animals and interact differently with predators. Indeed, behavioral assays involving prey, predators and *Nematostella* are consistent with this hypothesis. Further, the results of this work suggest a much wider and dynamic venom landscape than initially appreciated in animals with a complex life cycle.

DOI: https://doi.org/10.7554/eLife.35014.001

*For correspondence:
yehu.moran@mail.huji.ac.il

[†]These authors contributed equally to this work

**Competing interests:** The authors declare that no competing interests exist.

## Introduction

Venoms and the toxins they include are mostly used by animals for antagonistic interactions, such as prey capture and defense from predators (*Fry et al., 2009*; *Casewell et al., 2013*). Pharmacological research has been focused almost exclusively on the venoms of the adult stages despite the fact that many animals display remarkable transformations in body architectures and ecology during their development (*Ruppert et al., 2004*). Such vast differences may dictate different interspecific interactions for distinct life stages (*Wilbur, 1980*). As venom is hypothesized to be metabolically expensive, and in many cases highly specific (*Nisani et al., 2012*; *Casewell et al., 2013*), it is plausible that its composition might change between different life stages. Indeed, some ontogenetic variation was reported in the venoms of snakes (*Gibbs et al., 2011*), spiders (*Santana et al., 2017*) and cone snails (*Safavi-Hemami et al., 2011*), but until now this phenomenon was not studied thoroughly in an animal with a complex life cycle throughout its development.

The oldest extant group of venomous animals is the marine phylum Cnidaria, which includes sea anemones, corals, jellyfish and hydroids. Cnidarians are typified by their stinging cell, the nematocyte, that harbors a unique and highly complex organelle, the nematocyst (*Kass-Simon and Scappaticci, 2002*; *David et al., 2008*). This proteinaceous organelle is utilized as a miniature venom delivery system (*Thomason, 1991*; *Lotan et al., 1995*). Most cnidarians have a complex life cycle that includes both sessile benthic and mobile pelagic life stages of wide size distribution and ecological interactions. For example, the canonical life cycle of anthozoans (corals and sea anemones)

**eLife digest** Some animals produce a mixture of toxins, commonly known as venom, to protect themselves from predators and catch prey. Cnidarians – a group of animals that includes sea anemones, jellyfish and corals – have stinging cells on their tentacles that inject venom into the animals they touch.

The sea anemone *Nematostella* goes through a complex life cycle. *Nematostella* start out life in eggs. They then become swimming larvae, barely visible to the naked eye, that do not feed. Adult *Nematostella* are cylindrical, stationary 'polyps' that are several inches long. They use tentacles at the end of their tube-like bodies to capture small aquatic animals. Sea anemones therefore change how they interact with predators and prey at different stages of their life. Most research on venomous animals focuses on adults, so until now it was not clear whether the venom changes along their maturation.

Columbus-Shenkar, Sachkova et al. genetically modified *Nematostella* so that the cells that produce distinct venom components were labeled with different fluorescent markers. The composition of the venom could then be linked to how the anemones interacted with their fish and shrimp predators at each life stage.

The results of the experiments showed that *Nematostella* mothers pass on a toxin to their eggs that makes them unpalatable to predators. Larvae then produce high levels of other toxins that allow them to incapacitate or kill potential predators. Adults have a different mix of toxins that likely help them capture prey.

Venom is often studied because the compounds it contains have the potential to be developed into new drugs. The jellyfish and coral relatives of *Nematostella* may also produce different venoms at different life stages. This means that there are likely to be many toxins that we have not yet identified in these animals. As some jellyfish venoms are very active on humans and reef corals have a pivotal role in ocean ecology, further research into the venoms produced at different life stages could help us to understand and preserve marine ecosystems, as well as having medical benefits.
DOI: https://doi.org/10.7554/eLife.35014.002

includes a swimming larval stage (planula) that metamorphoses into a sessile polyp stage that matures into the reproductive adult.

The starlet sea anemone, *Nematostella vectensis,* is becoming a leading cnidarian lab model as unlike many other cnidarian species it can be grown in the lab throughout its life cycle. This makes *Nematostella* a unique system to study the venom of an animal with a complex life cycle. Another advantage is that the high genetic homogeneity of the common *Nematostella* lab strain minimizes individual genetic variation, which is far from trivial in most other venomous animals collected from the wild in limited numbers. Further, *Nematostella* has a sequenced genome and stage-specific transcriptomes and various molecular tools are available for its genetic manipulation (*Wikramanayake et al., 2003*; *Putnam et al., 2007*; *Renfer et al., 2010*; *Helm et al., 2013*; *Layden et al., 2016*). The *Nematostella* experimental toolbox is unique, not only for cnidarians, but also for venomous animals in general.

During the *Nematostella* life cycle, females release a gelatinous egg package and males release sperm into the water (*Hand and Uhlinger, 1992*). After fertilization occurs, the zygote cleavage begins, forming a blastula and less than 24 hr post-fertilization (hpf) gastrulation is completed. A planula larva emerges from the egg package 48–72 hpf and starts swimming in the water. Six to seven days after fertilization, the planula settles in soft substrate and starts to metamorphose into a primary polyp and sexual maturation takes about 4 months under lab conditions (*Hand and Uhlinger, 1992*) (*Figure 1A*). Whereas the egg and planula are roughly spherical and measure only about 250 µm, the morphologically differentiated elongated adult polyp can reach the length of 4 cm in the wild and up to 20 cm in the lab (*Williams, 1975*; *Hand and Uhlinger, 1992*). Venom is produced in *Nematostella* by ectodermal gland cells and nematocytes (*Moran et al., 2012b*; *Moran et al., 2013*). Although prey capture and production of Nv1, a major sodium channel modulator toxin (*Moran et al., 2008a*; *Moran et al., 2012b*), begins at the sessile primary polyp stage, nematocysts appear as early as 48 hpf in the swimming planula (*Marlow et al., 2009*). These are strong

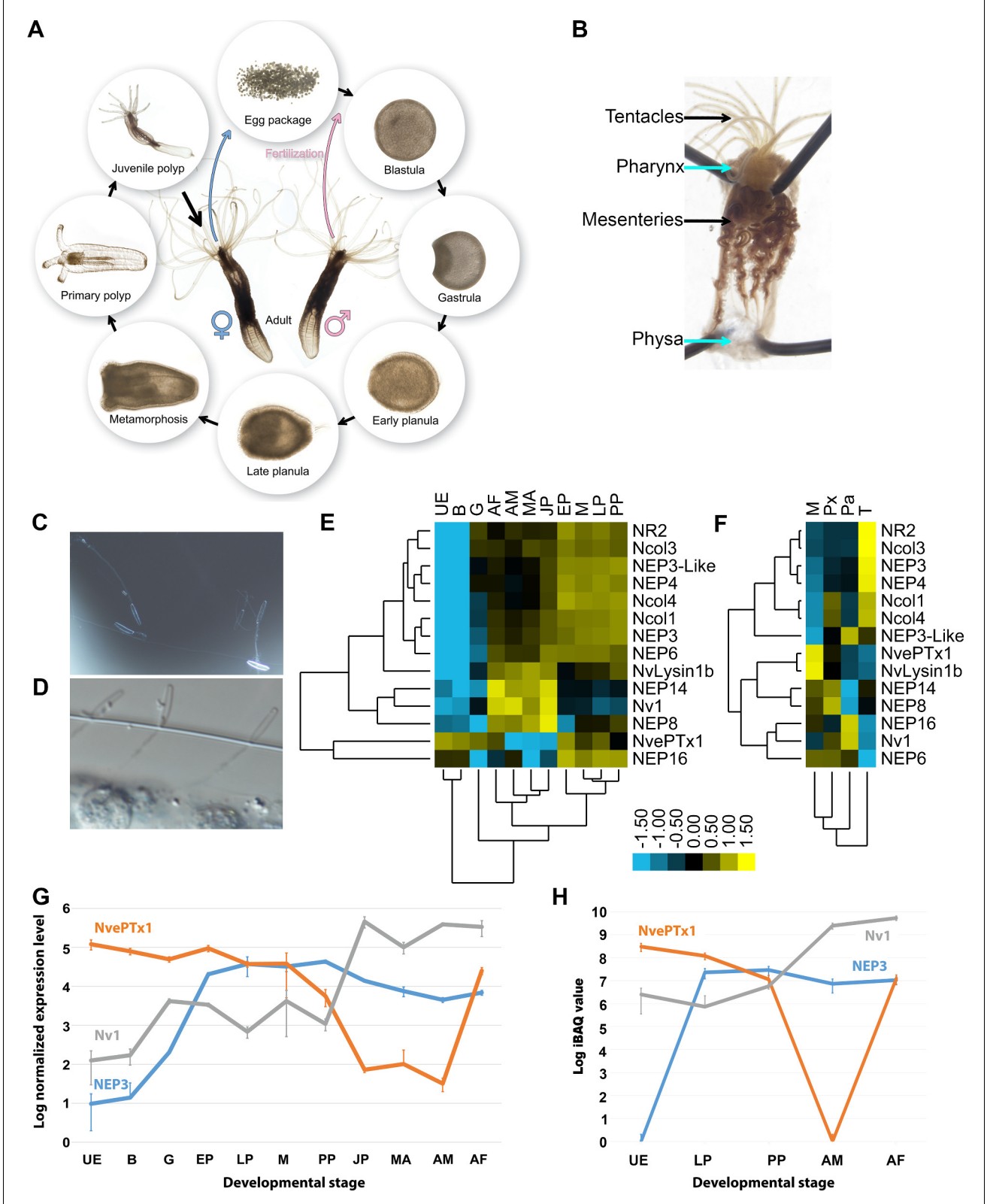

**Figure 1.** Expression of toxins across *Nematostella* development and tissues. (A) The life cycle of *N. vectensis*. (B) Dissected *Nematostella* female polyp. (C) Discharged planula nematocysts found in the medium after an encounter with *A. salina* nauplii. (D) Nematocysts of *Nematostella* planula pinned in the cuticle of an *A. salina* nauplii. (E–F) Heat maps of the nCounter differential expression levels of genes encoding toxins and other nematocyst proteins in various developmental stages and adult female tissues. (G) A graph at a logarithmic scale of the nCounter normalized

*Figure 1 continued on next page*

*Figure 1 continued*

expression levels of the genes encoding NEP3, NvePTx1 and Nv1 at each developmental stage. Each point is the average of three biological replicates and the error bars represent standard deviation. (H) A graph at a logarithmic scale of the mass spectrometry-measured iBAQ (Intensity-Based Absolute Quantification) values of the peptides NEP3, NvePTx1 and Nv1 at five developmental stages. Each point is the average of four technical replicates and the error bars represent standard deviation. Key for panels E and G: UE = Unfertilized Egg; B = Blastula; G = Gastrula; EP = Early Planula; LP = Late Planula; M = metamorphosis; PP = Primary Polyp; JP = Juvenile Polyp; MA = Mixed Adults; AM = Adult Male; AF = Adult Female. Key for panel F: M = Mesenteries; Px = Pharynx; Pa = Physa; T = Tentacles.

DOI: https://doi.org/10.7554/eLife.35014.003

indications that venom is likely present already in early life stages and its composition might dramatically change across the *Nematostella* life cycle. Further complexity in the regulation of venom production may be due to *Nematostella* producing different cell types that are unevenly distributed between various tissues (*Moran et al., 2013*) (*Figure 1B*).

Using an integrated and comparative approach, we carefully quantify and characterize the spatio-temporal expression of known and novel *Nematostella* toxins across different developmental stages and employ transgenesis to understand the dynamics of venom production in a species with a complex life cycle. We correlate the dynamics of toxin expression in life stages with organismal-level behavior experiments that highlight differential responses of prey and predators when exposed to these stages.

## Results

### *Nematostella* larvae are venomous

To assay the venomous potential of *Nematostella* larvae we incubated 4 days old swimming planulae with nauplii of the brine shrimp *Artemia salina*. Strikingly, within 10 min from the start of the incubation 3 out of 8 *Artemia* were paralyzed or dead, and within 90 min 7 of 8 were dead (*Video 1*), whereas in a control group without planulae all *Artemia* were alive. This experiment revealed that *Nematostella* planulae are capable of rapidly killing a crustacean that is larger than themselves. The relatively rapid effect and the size difference suggest that venom is involved in the process. Numerous discharged nematocysts were found in the water around the dead nauplii as well as in their cuticle (*Figure 1C–D*), further suggesting that the stinging capsules are involved in the envenomation process.

### Distinct and dynamic expression patterns of toxin genes in *Nematostella*

To accurately measure the expression levels of known toxin genes, putative toxins and genes encoding nematocyst structural proteins we used the medium-throughput nCounter platform (see materials and methods), which was previously shown to exhibit high sensitivity and precision similar to that of real-time quantitative PCR (*Prokopec et al., 2013*). We assayed the RNA expression levels of the genes encoding the sodium channel modulator neurotoxin Nv1 (*Moran et al., 2008a*), the putative toxins NvePTx1, NEP3, NEP3-like, NEP4, NEP8 and NEP16 (*Moran et al., 2013*; *Orts et al., 2013*), the putative metallopeptidases NEP6 and NEP14 (*Moran et al., 2013*), NvLysin1b, a cytolytic toxin which may also serve for food digestion (*Moran et al., 2012a*), and the structural components of the nematocyst capsule Ncol1, Ncol3 and Ncol4 (*David et al., 2008*; *Zenkert et al., 2011*), as well as the putative nematocyst structural component NR2 (*Moran et al., 2014*). The RNA measurements were performed on nine developmental stages (*Figure 1E*), adults of each sex and four dissected tissues of an adult female (*Figure 1F*; *Supplementary file 1*). The nCounter analysis revealed that many of the genes form informative clusters (*Figure 1E–F*). It is noticeable that the expression patterns of NEP3, NEP3-like and NEP4 strongly clustered with those of genes encoding structural nematocyst components in both the developmental and tissue analyses (*Figure 1E–F*), which is consistent with the finding that these putative toxins are produced in nematocytes and are released from the capsule upon discharge (*Moran et al., 2013*). Other toxins such as Nv1, which was shown to be expressed in polyp ectodermal gland cells (*Moran et al., 2012b*), or Nvlysin1b, produced by large gland cells in the pharynx and mesenteries beginning in early developmental stages

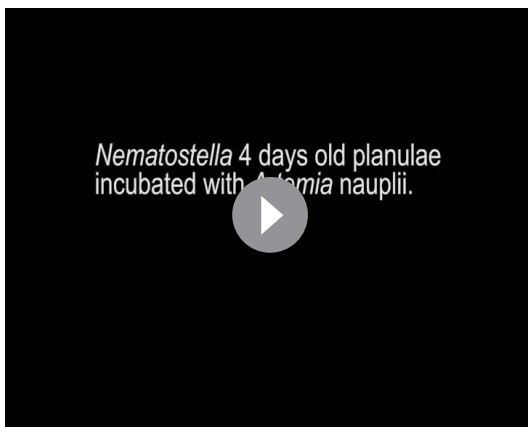

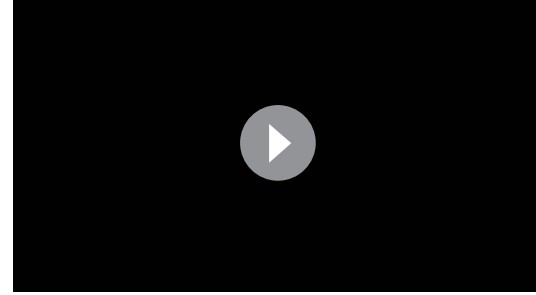

**Video 1.** The interaction between *Nematostella* planulae and *Artemia* nauplii. The panulae are rapidly paralyzing the nauplii after physical interaction. This movie is related to *Figure 1*.
DOI: https://doi.org/10.7554/eLife.35014.004

**Video 2.** The interaction between a burrowed adult *Nematostella* with grass shrimps. Only the *Nematostella* tentacles are exposed and can be encountered by the shrimp. The shrimp readily jumps away once it comes into contact with the tentacles.
DOI: https://doi.org/10.7554/eLife.35014.012

(*Moran et al., 2012a*), had different expression patterns in the nCounter analysis and did not form large clusters (*Figure 1E–F*). Expression levels of *NEP3*, *Nv1* and *NvePTx1* were strikingly distinct across development (*Figure 1G*). The expression of *Nv1* is relatively low in early developmental stages and then sharply peaks in the juvenile and adult polyps to extraordinary levels that are higher by almost two orders of magnitude compared to the other toxins (*Figure 1G*). These expression levels can be explained by the fact that the *Nematostella* genome contains more than a dozen gene copies encoding Nv1 (*Moran et al., 2008b*) and the peak in transcriptional expression late in development is consistent with earlier observations at the protein level (*Moran et al., 2012b*). In contrast to *Nv1*, *NEP3* is expressed at high levels already at gastrulation, peaks in the early planula and remains roughly stable throughout the rest of development into adulthood. Unlike the other toxins, the expression of *NvePTx1* peaks at the unfertilized egg, drops sharply across development and rises again in the adult female (*Figure 1G*).

As significant variations between toxin expression at the transcriptional level and protein level were recently reported (*Madio et al., 2017*), we complemented the expression of toxin genes in *Nematostella* using shotgun proteomics (tandem mass spectrometry MS/MS). We performed these quantifications in four replicates on lysates from four distinct developmental stages, as well as the separate sexes of the adult polyps (*Figure 1H*; *Supplementary file 2*). The dynamic expression patterns we observed at the protein level correlated well with the dynamics we observe at the transcriptomic level.

## NvePTx1 is a maternally deposited toxin

NvePTx1 was originally identified as a homolog of the type five potassium channel blocker BCsTx3 from the sea anemone *Bunodosoma caissarum* (*Orts et al., 2013*). We also identified bioinformatically several homologous sequences in the sea anemones *Anthopleura elegantissima* and *Metridium senile*, and the hydrozoan *Hydractinia symbiolongicarpus* (*Figure 2A*). This suggests that this peptide family was already present in the last common ancestor of all Cnidaria but was lost multiple times in various cnidarian lineages.

First, to test if NvePTx1 is indeed a toxin we expressed it in a recombinant form and incubated 20 zebrafish (*Danio rerio*) larvae with 0.5 mg/ml of highly pure recombinant peptide (assayed by reverse phase chromatography) for 20 hr. Upon the addition of the toxin, the fish reacted rapidly, with an increase in swimming speed. After 2 hr of incubation 10 fish larvae had died, with the remaining fish dying over the next 18 hr (at which point the experiment had ended) (*Figure 2B*). The control group (incubated in 5 mg/ml bovine serum albumin) behaved normally throughout the

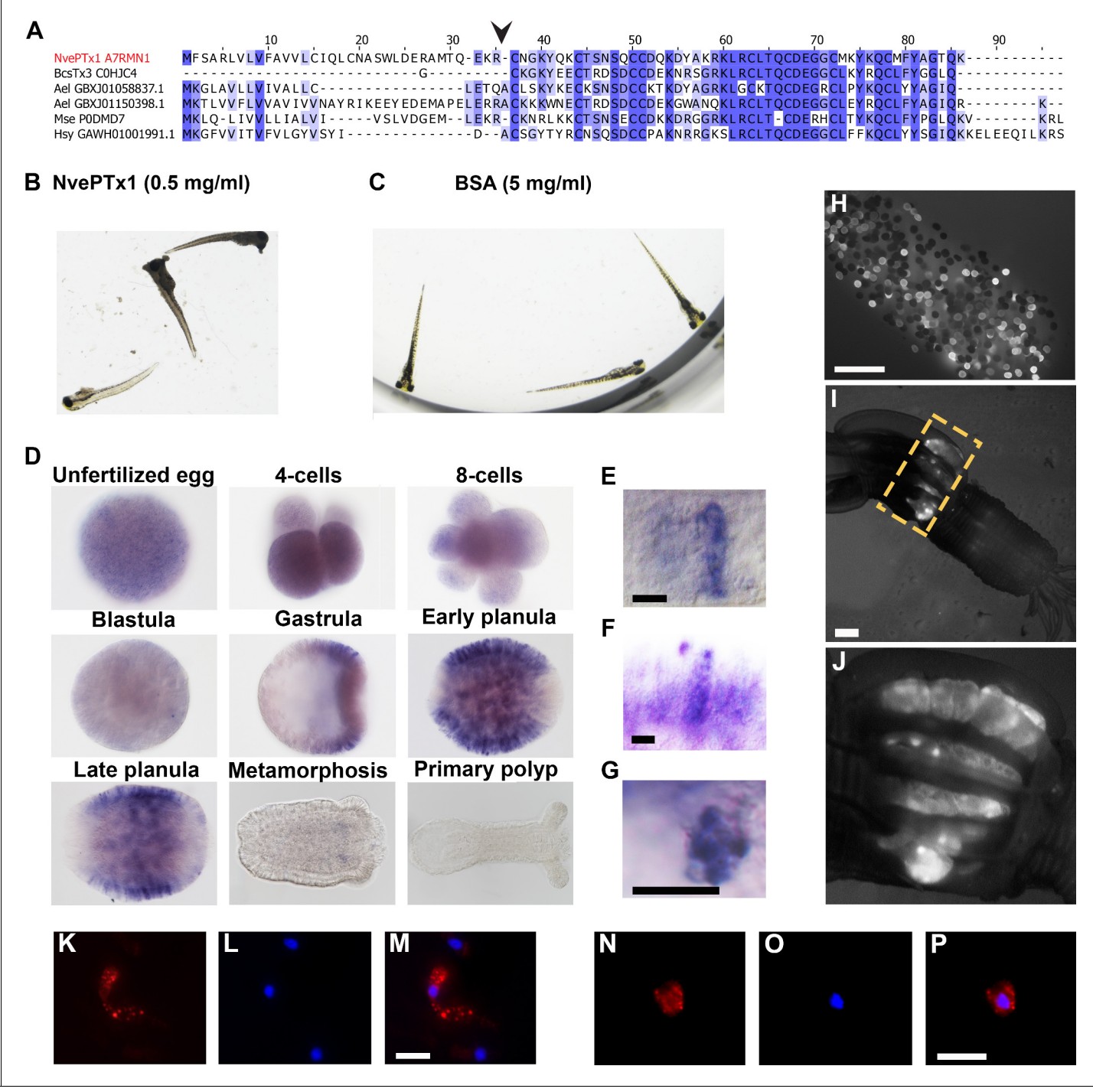

**Figure 2.** NvePTx1 is a toxin that is maternally deposited in *Nematostella* eggs. (A) Sequence alignment of NvePTx1 and its homologs from other cnidarian species. The arrowhead represents the proteolysis site that separates the mature toxin from the signal peptide. Accession numbers appear near each sequence. Nve = *Nematostella vectensis*; Bcs = *Bunodosoma caissarum*; Ael = *Anthopleura elegantissima*; Mse = *Metridium senile*; Hsy = *Hydractinia symbiolongicurpus*. (B–C) Zebrafish larvae 16 hr after incubation in NvePTx1 and BSA control. (D) Detection of *NvePTx1* expression by ISH. From the blastula stage on, the oral pole is to the right. Close up on planula large elongated (E–F) and small round (G) gland cells stained by ISH for *NvePTx1*. Gland cells, which are discernible by their round vesicles are free of stain, scale bar is 10 μm. (H–I) An adult F0 female polyp that as a zygote was injected by an *NvePTx1::mOrange2* construct. mOrange2 expression is noticeable in the gonads. Panel G is a close up of the region indicated in panel *H* by the yellow dotted box. Scale bar is 1000 μm (J) An egg package laid by the female from panel *H*. Many of eggs express mOrange2. Scale bar is 1000 μm. (K–M) Large elongated and (N–P) small round gland cells stained by anti mCherry. Scale bar is 10 μm.

DOI: https://doi.org/10.7554/eLife.35014.005

experiment and all larvae were alive after a 20-hr incubation (*Figure 2C*). To complement the nCounter experiment, we assayed the spatiotemporal expression pattern of *NvePTx1* by in situ hybridization (ISH). We observed that while *NvePTx1* is expressed uniformly throughout the unfertilized egg and early post-fertilization stages, in the gastrula the expression becomes spatially-localized and seems to be absent from the oral and aboral poles (*Figure 2D*). In the planula, the expression is clearly observed in the ectoderm in packed gland cells absent from the two body poles, and upon metamorphosis, the expression diminishes (*Figure 2D–E*). The ISH reveals two types of gland cells, one large and elongated (*Figure 2E–F*) and another small and round (*Figure 2G*). The results of the ISH and nCounter experiments indicated that *NvePTx1* is maternally deposited at the RNA level. Further, we could detect NvePTx1 peptide hits in our MS/MS analyses data (*Figure 1H*) as well as in supplementary datasets available from previous proteomic studies of *Nematostella* eggs (*Lotan et al., 2014*; *Levitan et al., 2015*), suggesting maternal deposition also at the protein level. To directly test this, we have injected into *Nematostella* zygotes a transgenesis construct that carries the gene encoding the fluorescent reporter mOrange2 (*Shaner et al., 2004*) fused to an NvePTx1 signal peptide downstream of a putative *NvePTx1* promoter. Noticeably, several females of the injected first generation (F0) exhibited strong expression of mOrange2 in round structures in their mesenteries, which are most probably the ovaries where the eggs are formed (*Figure 2H–I*). This observation is congruent with the fact that the mesentery is the only female adult tissue where high *NvePTx1* transcript levels are detected at high levels by the nCounter analysis (*Figure 1F*). Further, upon induction of spawning, the female polyps with the fluorescent mesenterial tissue released egg packages harboring plenty of fluorescent eggs (*Figure 2J*), strongly supporting maternal deposition of NvePTx1. Next, by performing an immunostaining assay on F1 transgenic animals we were able to verify the presence of two distinct types of ectodermal gland cells as identified by the ISH results (*Figure 2K–M*).

## Phylogeny, primary structure and activity of NEP3

The *NEP3*, *NEP4*, and *NEP8* gene products were previously detected by MS/MS to be released from *Nematostella* nematocysts upon discharge and hence were hypothesized to be putative toxins (*Moran et al., 2013*). We detected an additional gene encoding a NEP3 homolog in the *Nematostella* genome and named it *NEP3-like*. The four nucleotide sequences were translated in silico and were suggested to encode precursors of secretory proteins equipped with typical signal peptides (according to SignalP online tool) (*Petersen et al., 2011*). An additional search in the Pfam database (*Finn et al., 2010*) showed that each precursor contains three ShKT sequence motifs (PF01549, *Figure 3A*) typical for several potent cnidarian toxins (*Aneiros et al., 1993*; *Castañeda and Harvey, 2009*). Based on these common features and their sequence similarity, we designated *NEP3*, *NEP3-like*, *NEP4*, and *NEP8* as the 'NEP3 family'. We searched publicly available transcriptomic shotgun assembly databases and identified several sequences of homologs from the sea anemones *Edwardsiella lineata*, *Aiptasia pallida* and *Anthopleura elegantissima* as well as the stony corals *Acropora digitifera* and *Stylophora pistillata*, showing significant sequence similarity and identical domain structure to the NEP3 family members in *Nematostella* (*Figure 3—figure supplement 1*). A phylogenetic analysis revealed that each of NEP3, NEP4 and NEP8 from *Nematostella* formed a strongly supported (bootstrap values > 0.5) clade with a highly similar protein from *Edwardsiella* (*Figure 3B*), indicating orthology. Thus, the new sequences from *Edwardsiella* were named according to their *Nematostella* orthologs. As *Nematostella* and *Edwardsiella* are members of the basally branching sea anemone family Edwardsiidae (*Rodríguez et al., 2014*; *Stefanik et al., 2014*) and possess NEP3, NEP4 and NEP8, we can infer that those three proteins probably originated in the last common ancestor of the Edwardsiidae. Other cnidarian species bear more distantly related NEP3 family members, and their exact orthologous or paralogous nature could not be determined due to low bootstrap values (*Figure 3B*). However, their presence in stony corals suggests that those proteins already appeared 500 million years ago in the last common ancestor of stony corals and sea anemones (*Shinzato et al., 2011*), but were lost in multiple hexacorallian lineages.

To check whether NEP3 serves as a toxin, we planned to express it in a recombinant form. However, it was initially not clear in what native form this protein is found in the animal as we detected a potential Lys-Arg tandem, which is a prominent cleavage signal in nematocyst proteins (*Anderluh et al., 2000*), between the first and second domains of NEP3. Hence, we decided to first explore the primary structure of the native mature NEP3. For this aim, we discharged lyophilized

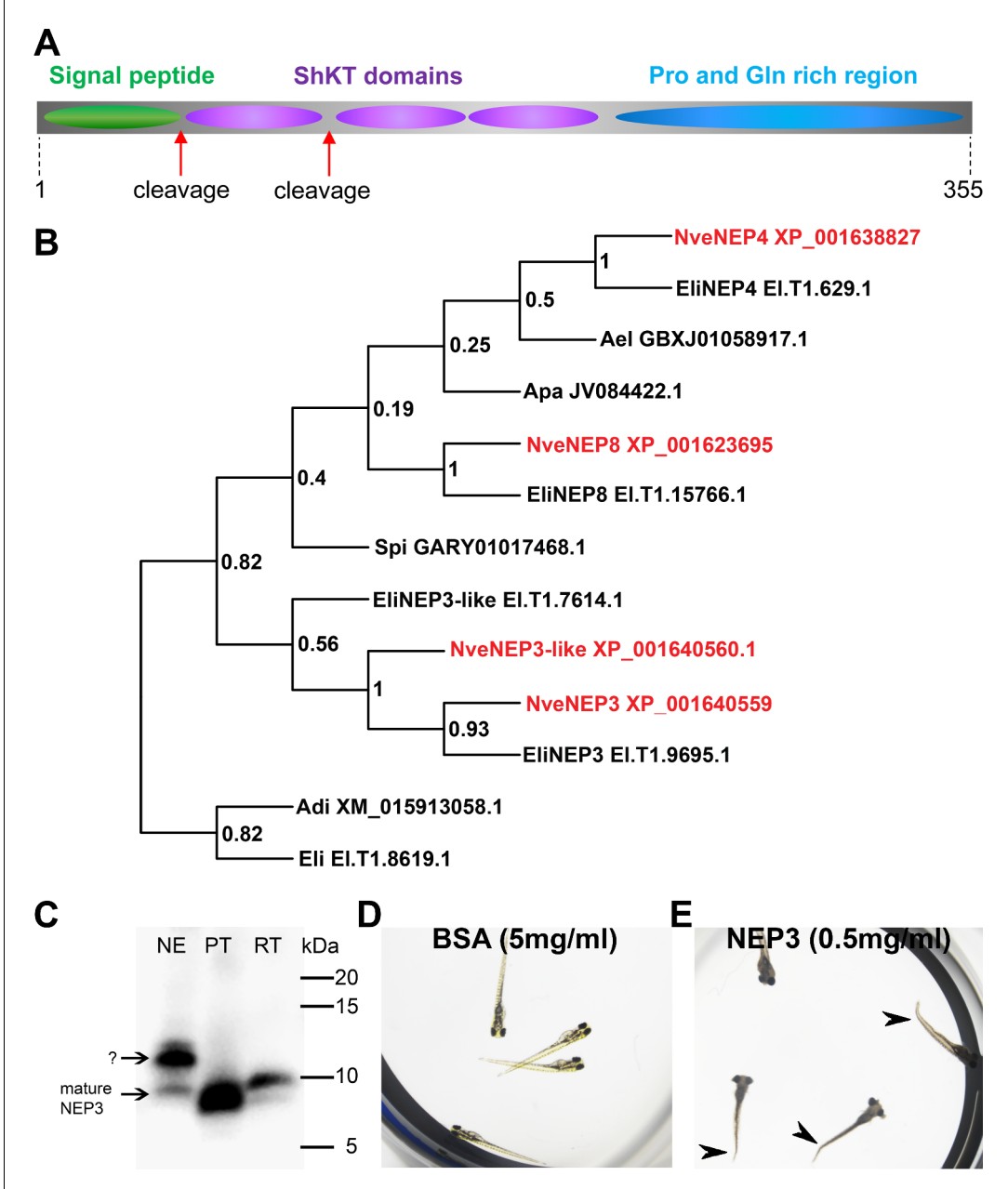

**Figure 3.** NEP3 is a toxin that is processed from a long precursor protein and is part of a large family. (**A**) The primary structure of the NEP3 precursor. (**B**) A maximum-likelihood tree of the NEP3 family. Accession numbers appear near each sequence and bootstrap values (fraction of 1000 bootstraps) appear near each node. Nve = *Nematostella vectensis*; Eli: *Edwardsiella lineata*; Ael = *Anthopleura elegans*; Apa = *Aiptasia pallida*; Spi = *Stylophora pistillata*; Adi = *Acropora digitifera*. (**C**) Western blot with rat Anti-NEP3 antibody. Samples are discharged nematocyst extract (NE), native purified toxin (PT) and recombinant toxin (RT). (**D–E**) Zebrafish larvae after 16 hr incubation in NEP3 and BSA control. Arrowheads are pointing at twitched tails.
DOI: https://doi.org/10.7554/eLife.35014.006

The following figure supplement is available for figure 3:

**Figure supplement 1.** Multiple sequence alignment of the NEP3 family members from *Figure 3B*.
DOI: https://doi.org/10.7554/eLife.35014.007

nematocysts and analyzed the molecular weight of NEP3 making part of the ejected protein mixture by western blot with custom polyclonal antibodies against the first NEP3 ShKT domain. However, the western blot resulted in two major bands (~10 and 12 kDa) (*Figure 3C*). A gradual three-step FPLC procedure of gel filtration, anion exchange and reverse phase chromatography was applied to

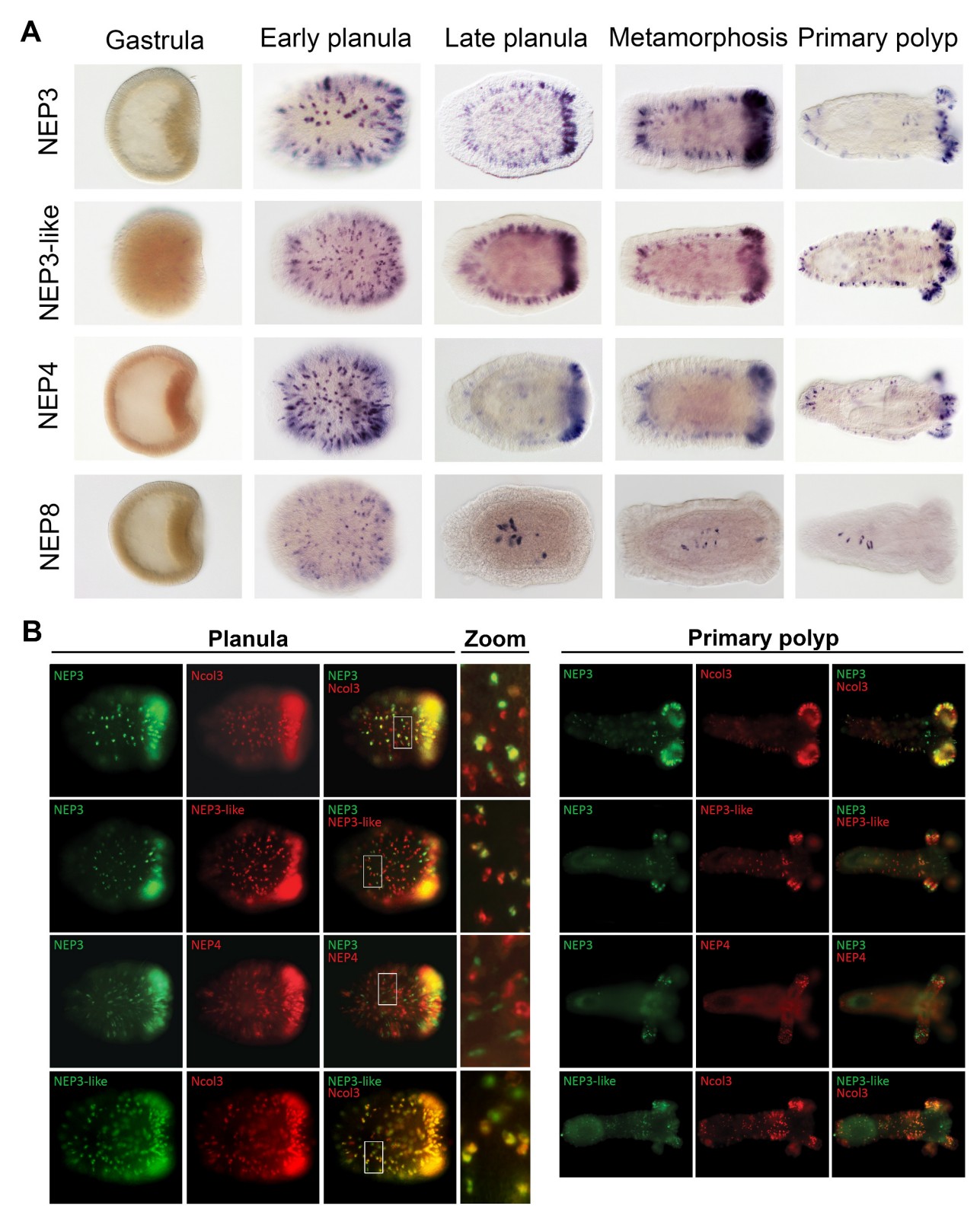

**Figure 4.** Partially overlapping and distinct expression patterns of NEP3 family members. (**A**) Expression of the NEP3 family members in five developmental stages of *Nematostella* as determined by ISH. (**B**) Expression of the NEP3 family members in planulae and polyps of *Nematostella* as determined by dFISH. In all panels, the oral pole is to the right.

DOI: https://doi.org/10.7554/eLife.35014.008

*Figure 4 continued on next page*

*Figure 4 continued*

The following figure supplement is available for figure 4:

**Figure supplement 1.** Close up on ISH staining of *NEP3*, *NEP3-like*, *NEP4* and *NEP8* reveals that they are expressed in nematocytes.
DOI: https://doi.org/10.7554/eLife.35014.009

purify the dominant NEP3 fragment, and at each stage we carried on with the fraction with the strongest western blot signal. This procedure yielded a nearly pure fraction of mature NEP3 fragment with molecular weight ~10 kDa. Comparison of the electrophoretic mobility of the native mature NEP3 and a recombinant peptide corresponding to the first domain showed that the two peptides possess highly similar molecular weights (*Figure 3C*). Thus, we conclude that a native peptide composed of only the first domain of NEP3 is found in *Nematostella* and is released from the nematocyst upon discharge. Following this finding we carried out a toxicity test where we incubated zebrafish larvae with 0.5 mg/ml recombinant mature NEP3. While in the control group (incubated for 20 hr in 5 mg/ml bovine serum albumin) all the 20 fish survived (*Figure 3D*), all 17 *Danio* fish larvae in the NEP3-treated group died within 5 hr and the larvae exhibited pronounced contraction and tail twitching (*Figure 3E*) suggesting that the mature NEP3 peptide might be neurotoxic.

## Different nematocytes express different NEP3 family members

To gain improved resolution of the expression of the four NEP3 family members, we employed ISH to localize their expression in five developmental stages. All four genes are expressed in nematocytes on the surface of the early planula (*Figure 4A*; *Figure 4—figure supplement 1*). However, beginning at the late planula stage the expression of *NEP8* shifts to only a handful of nematocytes in the lower pharynx (*Figure 4A*). In contrast to *NEP8*, the three other family members, *NEP3*, *NEP3-like* and *NEP4* are expressed throughout the ectoderm in nematocytes, with high concentration of expressing cells in the oral pole. In the primary polyp, expression of *NEP3*, *NEP3-like* and *NEP4* is noticeable in nematocytes in the body wall and physa ectoderm and in the upper and lower pharynx (*Figure 4A*). Further, in tentacle tips, which are very rich with nematocytes, there are large numbers of nematocytes expressing the three toxins, fitting well our nCounter results (*Figure 1F*). A superficial look on these results may give the impression that in *Nematostella* there is one small population of pharyngeal nematocytes that express *NEP8* and another very large population of nematocytes that express *NEP3*, *NEP3-like* and *NEP4* in much of the ectoderm. However, we decided to use double fluorescent ISH (dFISH) to check if this is truly the case.

In the dFISH experiments, we localized the mRNA combinations of *NEP3* with *Ncol3*, *NEP3* with *NEP3-like*, *NEP3* with *NEP4*, and *NEP3-like* with *Ncol3*. Unexpectedly, all the combinations showed only limited overlap in their expression, with *NEP3-like* and *Ncol3* showing the highest overlap and *NEP3* and *NEP4* showing the lowest (*Figure 4B*). This result can be explained by two non-exclusive explanations: the first is that the nematogenesis (production nematocysts) is a highly dynamic process that requires different genes to be expressed at different times along the nematocyst maturation process; the second is that the three family members are mostly expressed in different nematocyte populations and only few nematocytes express all family members.

To test the latter hypothesis, we injected *Nematostella* zygotes with a construct carrying a chimera of the signal peptide of NEP3 with mOrange2 downstream of the putative promoter of *NEP3*. Zygotes started expressing mOrange2 in nematocytes about 4 days after injection. We raised the positive F0 animals to adulthood and then crossed them with wildtype polyps to identify founders. We found six female and three male founders that the mOrange2 expression in their F1 progeny imitated the expression of the native gene (*Figure 5A–C*). To verify this, we performed double ISH on F1 as well as F2 animals (third generation) and found nearly perfect overlap between the transcriptional expression of the mOrange2 transgene and that of the NEP3 toxin gene (*Figure 5D–F*). mOrange2 expression was observed in many nematocytes with especially dense populations in the tentacle tips and the mesenteries (*Figure 5C*). Strong expression of mOrange2 was also detected in numerous nematocytes within the nematosomes (*Figure 5G–I*), defensive structures that *Nematostella* releases to its surroundings and within egg packages (*Babonis et al., 2016*).

Next, we dissociated tentacles of transgenic F1 animals by a mixture of commercial proteases into single cells and observed what cells express mOrange2 and hence NEP3. As expected, we

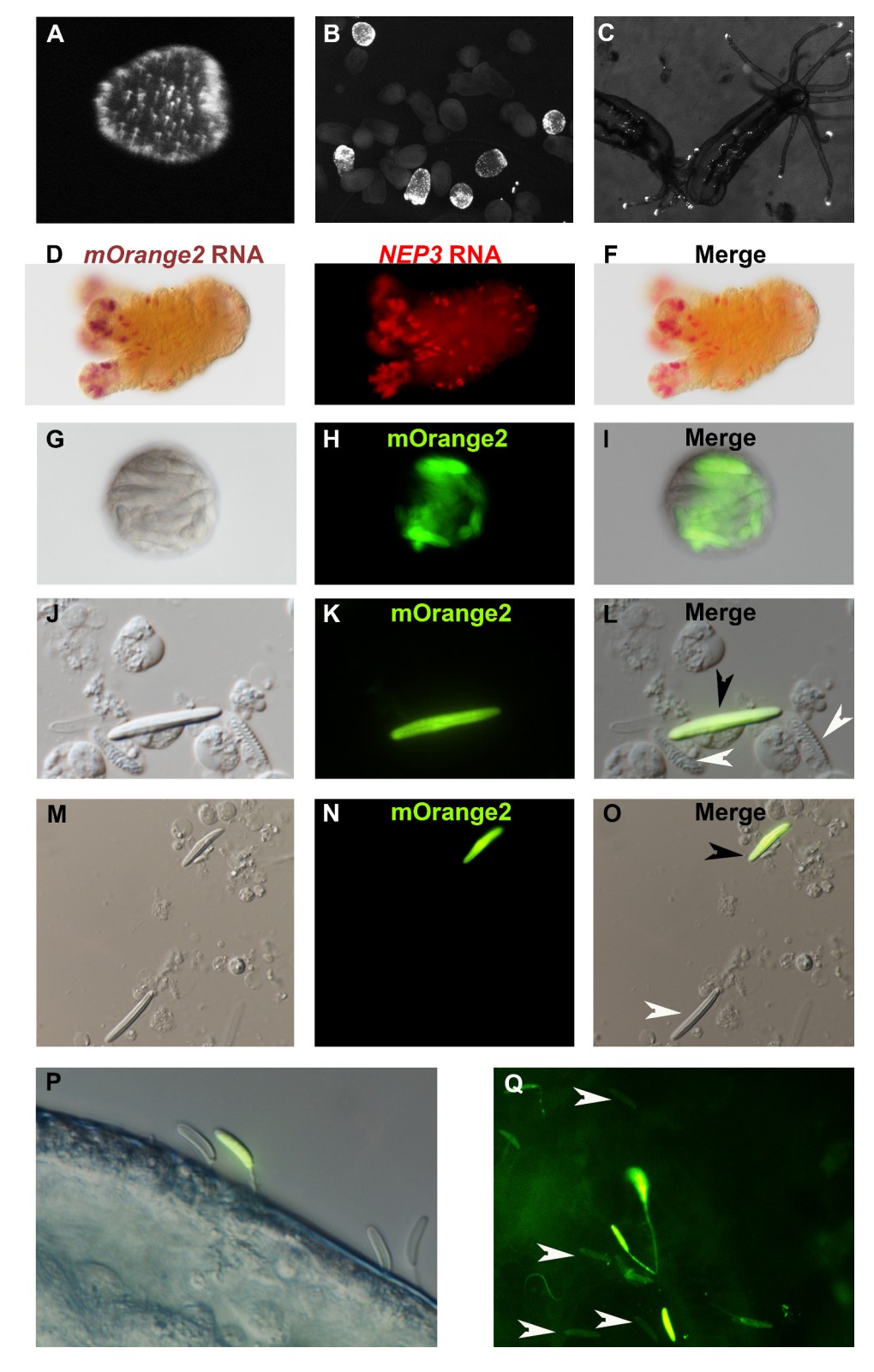

**Figure 5.** NEP3 is expressed only in a subset of nematocytes. (**A**) Three days of planula of an F1 transgenic line expressing mOrange2 under a *NEP3* promoter. (**B**) 6 days old metamorphosing larva of the same transgenic line. (**C**) Two months old juvenile polyps of the same line. (**D–F**) Double ISH of an F2 primary polyp of the transgenic line with probes against NEP3 and mOrange2. mOrange2 transcripts were stained with an NBT (nitro-blue tetrazolium chloride) and BCIP (5-bromo-4-chloro-3′-indolyphosphate p-toluidine) solution that forms purple crystals, whereas NEP3 transcripts were

*Figure 5 continued on next page*

*Figure 5 continued*

stained with FastRed that provides fluorescent red signal. (**G–I**) A nematosome of the transgenic line exhibiting mOrange2 fluorescent signal within its nematocytes. (**J–L**) A spread of dissociated cells of the tentacles of an F2 transgenic NEP3 polyp. An mOrange2-positive nematocyte is indicated by a black arrow and negative spirocytes are indicated by white arrows. (**M–O**) A picture of another field of the cell spread from previous panels. An mOrange2-positive nematocyte is indicated by a black arrow and a negative nematocyte is indicated by a white arrow. Scale bar is 10 μm in panels G-O. (**P**) Nematocytes of a transgenic F2 polyp pinned in the cuticle of *A. salina*. Only one nematocyte in the picture is mOrange2-positive. (**Q**) Nematocytes of an F2 polyp of the same transgenic line are pinned in the skin of a zebrafish larva. Only some of them are mOrange2 positive. White arrows indicate examples for negative nematocytes.

DOI: https://doi.org/10.7554/eLife.35014.010

observed that NEP3 is expressed in nematocytes, but not in spirocytes (*Figure 5J–L*), which are believed to be used for entangling prey and not for venom delivery (*Mariscal et al., 1977*). Moreover, we also observed that NEP3 is expressed only in a subpopulation of nematocytes (*Figure 5M–O*), suggesting like the ISH and dFISH experiments that different nematocytes express different toxins.

However, at that point there was a possibility that the mOrange2 is noticeable only in developing nematocytes and that mature capsules are not glowing due to various technical limitations such as the mature capsule wall obstructing light. In order to test whether there are mature mOrange2-positive capsules in our transgenic line, we challenged the F1 polyps with *Artemia* nauplii and zebrafish larvae. We then took the attacked prey items and visualized them with fluorescent microscopy. Strikingly, mOrange2-positive capsules with glowing tubules were pinned in the crustacean and fish cuticle or skin, respectively (*Figure 5P–Q*), indicating that those are mature capsules. However, these capsules were accompanied by other capsules that were mOrange2-negative, strongly suggesting once again that only a certain nematocyte subpopulation in *Nematostella* is expressing NEP3.

## Interaction between *Nematostella* and potential predators and prey

At different developmental stages *Nematostella* inhabits various ecological niches, and consequently its interaction with predators and prey may change throughout the life cycle. Here, we tested *Nematostella* interactions with the grass shrimp *Palaemonetes* sp. and the killifish *Fundulus heteroclitus* at egg, planula, primary polyp, and adult life stages (*Table 1*). Grass shrimps are reportedly predators of *Nematostella* (*Kneib, 1985*; *Kneib, 1988*), however, in our observations, when encountering the tentacles of adult polyps of *Nematostella* burrowed in substrate, shrimps immediately 'jumped' away from the tentacles (*Video 2*). In contrast, in the absence of the typical mud substrate, shrimps in an environment lacking food would consume adult polyps by starting to feed at the side of their body column, avoiding contact with the tentacles. These results are in agreement with a previous study that found that shrimps can generally prey on *Nematostella* polyps, but not when they are burrowed in the substrate (*Posey and Hines, 1991*). The response of grass shrimp to tentacle contact correlates well with the high sensitivity (LD$_{50}$ = 1.25 ng/100 mg shrimp) of this species to Nv1 toxin, which is highly expressed in tentacles starting from the juvenile polyp stage.

Our results do indicate that *Nematostella* may be a potential food source for grass shrimp at earlier developmental stages (disassociated eggs, egg packages, planulae larvae, and primary polyps). Grass shrimp kept in an environment without food for more than 2 days consumed disassociated eggs and egg packages, which may be due to young embryonic stages not expressing Nv1. Further,

**Table 1.** Interaction of *Nematostella vectensis* with potential predators, *Palaemonetes* sp. and *Fundulus heteroclitus*, at different life stages. ++ readily eats/eats when food is restricted for at least 2 days,+eats when food is restricted for at least 7 days, - does not consume or consider a food source, – attempted to consume, but exhibited an adverse reaction in all treatments, NA not available. *Only when the eggs are treated with cysteine.

| | Egg package | Individual eggs | Planulae | Primary polyps | Adults (body exposed) | Adults burrowed in the substrate |
|---|---|---|---|---|---|---|
| Grass shrimp | ++ | ++ | - | + | + | – |
| Fish larvae | – | ++* | – | – | – | - |
| Fish adults | - | - | - | + | - | NA |

DOI: https://doi.org/10.7554/eLife.35014.011

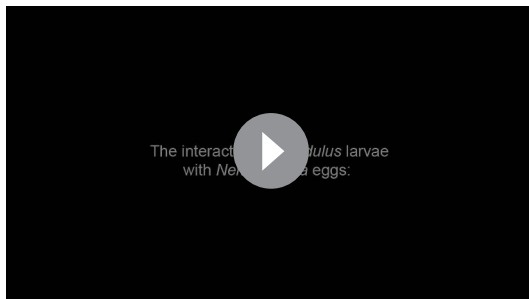

**Video 3.** The interaction of *Fundulus* larvae with *Nematostella* egg package. Upon contact with the egg package, *Fundulus* reacts swiftly and escapes.
DOI: https://doi.org/10.7554/eLife.35014.013

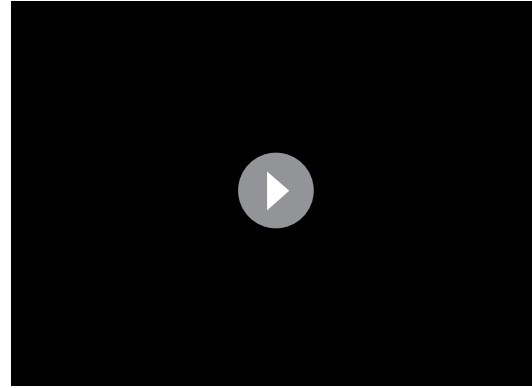

**Video 4.** The interaction of *Fundulus* larvae with *Nematostella* planulae. *Fundulus* actively expels planulae from their mouths, if they were erroneously perceived as food.
DOI: https://doi.org/10.7554/eLife.35014.014

injections of embryonically-expressed NvePTx1 and NEP3 (50 ng/100 mg shrimp) did not produce noticeable symptoms in grass shrimp. We did not observe any planulae larvae being consumed by grass shrimp. However, it is likely that grass shrimp do not regularly encounter planulae larvae as they have the potential to disperse away from the substrate in which the shrimp usually feed. When grass shrimps were provided with primary polyps they were consumed after 7 days when no other food sources were provided, suggesting that this food source is likely not preferred, but is ingested.

*Fundulus* is omnivorous, and was reported to feed on a large variety of benthic organisms (*James-Pirri et al., 2001*; *McMahon et al., 2005*) and possibly prey on *Nematostella* (*Wiltse et al., 1984*). In the lab, adult fish did not attempt to eat adult polyps, nor did they attempt to eat disassociated eggs or planulae larvae, which may be due to their small size relative to the adult fish and may not even be recognized as a potential food source. Surprisingly, the fish larvae did feed on eggs when separated from the gelatinous portion of the egg package by cysteine treatment, but ejected from their mouths eggs that were separated from the package mechanically (*Video 3*). This suggests that the eggs might carry defensive compounds that are removed or inactivated by the cysteine. Fish do not eat eggs if encased within the egg package and attempted to remove the gelatinous egg package from their tails following coincidental encounters (*Video 3*). When provided planulae as a prey item, *Fundulus* larvae attempted to swallow the larvae but immediately released them and swam away (*Video 4*).

In the lab, *Fundulus* larvae (1–3 days post hatching) were consumed by adult *Nematostella* without substrate, but managed to avoid predation by *Nematostella* when substrate was present, however, this is likely due to the added vertical space provided to the fish larvae within the dishes containing substrate and anemones. Given more time, a coincidental encounter may occur, resulting in predation of the *Fundulus* larvae. *Fundulus* larvae actively avoid primary polyps, however, adults will consume them when food is limited (*Table 1*). To test the significance of this observation we exposed fish larvae to three treatments: dead *Artemia* (food source), silica beads (inert treatment), and primary polyps. *Fundulus* larvae spent more time in the bottom of the aquarium than the top when silica beads or *Artemia* were present and spent more time at the top than the bottom when the primary polyps were present (*Figure 6*). Time spent at each location was significantly different in our one way ANOVA analysis for top (p value = 0.035) and bottom (p value = 0.01) for each location, with the Tukey analysis indicating that the significant difference was found between primary polyps and the other two treatments, but not when comparing silica beads and *Artemia* at both locations (*Figure 6*). Thus, we can conclude that *Fundulus* larvae tend to avoid interactions with *Nematostella* primary polyps.

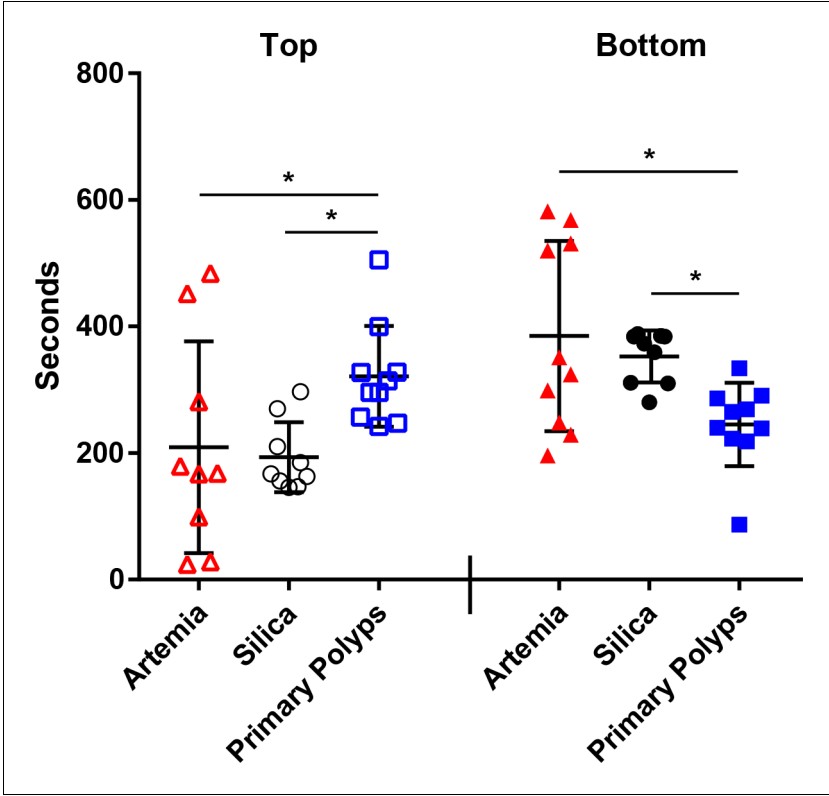

**Figure 6.** Time (seconds) each fish spent when exposed to *Artemia* (red), silica beads (black), or *Nematostella* primary polyps (blue) in the top (open symbols) or bottom (closed symbols) third of the aquarium. Statistically significant differences between treatments identified using separate one way ANOVAs and the Tukey post hoc test are noted with an asterisk.

DOI: https://doi.org/10.7554/eLife.35014.015

## Discussion

Our finding of different expression levels of toxins in different developmental stages and adult tissues strongly suggests that venom composition changes across development and that each arsenal of toxins might have been shaped by selection for different biotic interactions. As *Nematostella* develops from a non-predatory, swimming larva to an adult sessile predatory polyp that is 150-fold larger than the larva (*Figure 1A*), its interspecific interactions vastly change across development. For example, *Nematostella* egg packages can be consumed by grass shrimp, but it is highly unlikely that adult polyps are part of the grass shrimp diet (*Table 1*; *Video 2*). These observations correlate with the expression dynamics of Nv1 that is highly toxic for shrimps. Unlike the grass shrimp, *Fundulus* larvae do not consume *Nematostella* egg packages and actively expel from their mouths planulae, which were erroneously perceived as food (*Videos 3* and *4*). They consumed only individual eggs following cysteine washes, which do not represent native conditions. The reduced occurrence of predation can be explained by the presence of various toxins, such as members of NEP3 family, which may protect *Nematostella* from fish predators even at very early developmental stages. This notion is supported by the toxic effects of NEP3 on zebrafish larvae.

We hypothesize that these dynamic interactions, coupled with the potentially high metabolic cost of toxins (*Nisani et al., 2012*), have driven the evolution of a distinct venom composition in each developmental stage. Moreover, in the case of cnidarians, an additional metabolic cost stems from the fact that nematocysts are single-use venom delivery apparatuses that have to be reproduced in very high numbers after each antagonistic interaction. Hence, there is a clear advantage in using a highly adapted venom in each developmental stage.

Because venom in planulae is used purely for defense, whereas the venom in polyps is used both for defense and for prey capture, our results also relate to previous results in scorpions

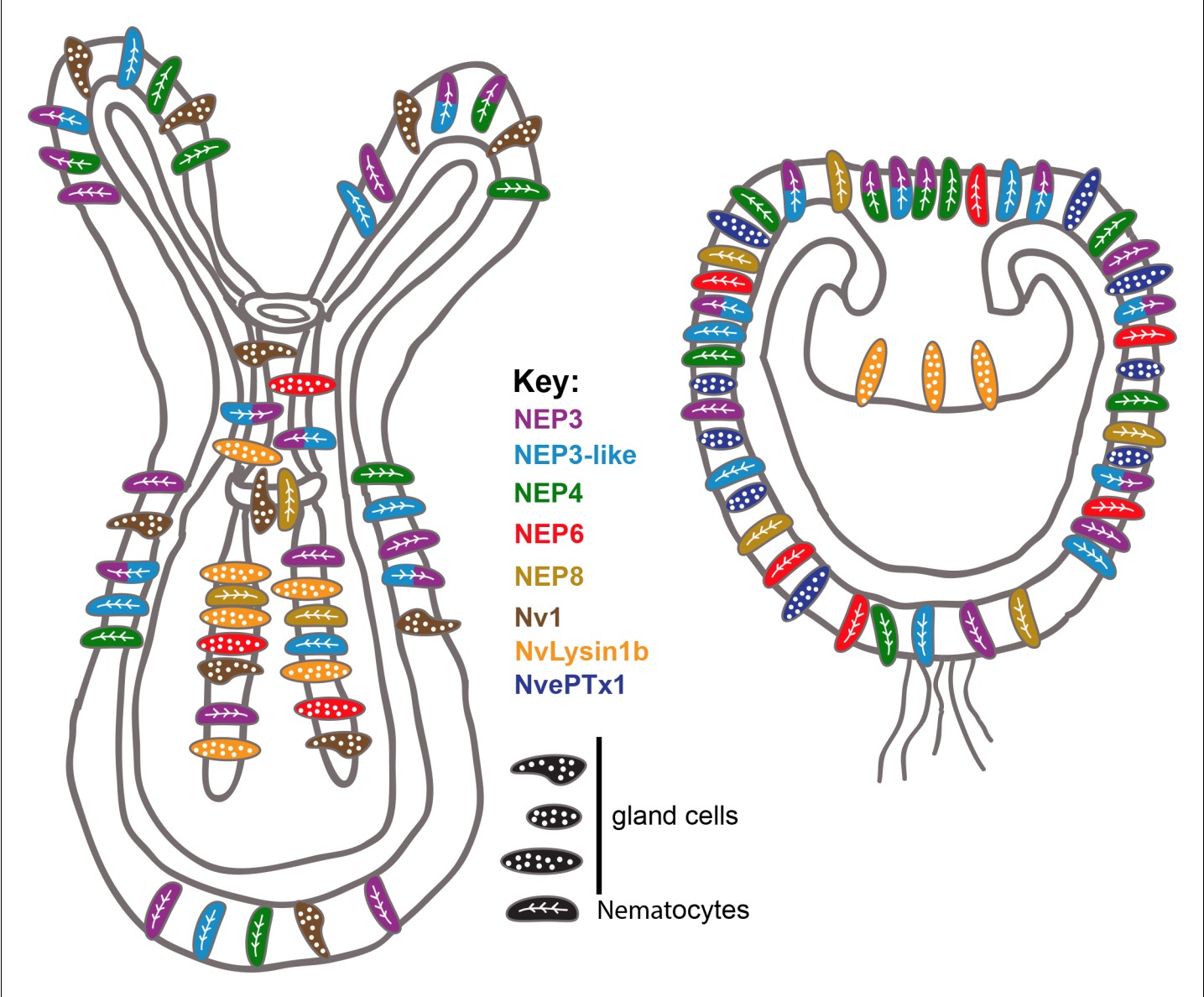

**Figure 7.** Summary of the current knowledge on spatiotemporal toxin expression in early planula and primary polyp of *Nematostella.* This illustration summarizes the current work as well as previous works (*Moran et al., 2012a*; *Moran et al., 2012b*; *Moran et al., 2013*). The degree of expression overlap of NEP8 with other toxins is currently unknown.

DOI: https://doi.org/10.7554/eLife.35014.016

(*Inceoglu et al., 2003*) and cone snails (*Dutertre et al., 2014*), where venom compositions for defense and prey-capture were shown to differ. Further, some of the toxins we localized can be attributed for specific functions based on their expression patterns. For example, it is very likely that NvePTx1 is a defensive toxin due to its occurrence in the egg and in ectodermal gland cells of the planula (*Figures 2D–H* and *7*), whereas NEP8 is probably used in the polyp for killing prey after it is swallowed, as it is expressed exclusively in the lower pharynx and mesenteries (*Figures 4A* and *7*).

Chemical protection of the eggs was reported in several animals such as black widow spiders (*Buffkin et al., 1971*; *Lei et al., 2015*), snails (*Dreon et al., 2013*), octopuses and some fishes and amphibians (*Bane et al., 2014*). Our data on *Nematostella* NvePTx1 expressed in eggs and embryonic stages further support the idea of ecological importance of chemical defense in early life stages across the animal kingdom.

Our findings that NEP3 family members are expressed in different population of nematocytes reveals that nematocyte diversity in *Nematostella* exceeds well-beyond the morphology-based assessments, which revealed only two types of nematocysts in *Nematostella*: basitrichous haplonemas (also called basitrichs) and microbasic mastigophores (*Frank and Bleakney, 1976*; *Zenkert et al., 2011*). A study in Hydra discovered that two members of a single pore-forming toxin family are expressed in two morphologically distinct types of nematocytes (*Hwang et al., 2007*), but to the best of our knowledge, similar complexity of toxin expression patterns in morphologically similar nematocytes was never reported before. Further, mechanisms of venom biosynthesis at exact cellular level resolution have not been reported in more complex venomous organisms as well. However, at lower resolution, some reports suggest that within one venom gland different secretory units are specialized on production of a limited number of toxins (*Dutertre et al., 2014*; *Undheim et al., 2015*). Thus, specialized venom secretory cells are probably a common trait among venomous animals.

The distinct expression patterns of the NEP3 family also provides important indications regarding toxin evolution. For example, the expression of *NEP8* in pharyngeal nematocytes, and its absence from the tentacles and outer body wall, where its paralogs *NEP3*, *NEP3-like* and *NEP4* are expressed (*Figures 4A* and *7*), is an indication for sub- or neo-functionalization. The specialization of the different family members is also supported by their conservation in *Edwardsiella* (*Figure 3B*).

Variation in expression patterns of the NEP3 family members and the fact that at least four different types of gland cells at distinct developmental stages and tissues express different toxins (Nv1, Nvlysin1b, NEP6 and NvePTx1) in *Nematostella* suggests a highly complex venom landscape in this species (*Figure 7*). At first glance, such a system might seem relatively inefficient. However, we hypothesize that harboring many different toxin-producing cell types, provides modularity and enables evolutionary plasticity of toxin expression. Indeed, our results as well as results of others, suggest that different sea anemones species express similar toxins in different cell types (*Moran et al., 2012b*) and different tissues (*Macrander et al., 2016*). This evolutionary plasticity might be one of the factors that made sea anemones such a successful group that inhabits all the world's oceans for the last 600 million years.

## Materials and methods

### Sea anemone culture

*Nematostella* embryos, larvae and juveniles were grown in 16‰ sea salt water at 22°C. Adults were grown in the same salinity but at 17°C. Polyps were fed with *Artemia* nauplii three times a week. Induction of gamete spawning was performed according to a published protocol (*Genikhovich and Technau, 2009b*).

### nCounter analysis

Total RNA from different developmental stages and body parts of adult female *Nematostella* was extracted with Tri-Reagent (Sigma-Aldrich, St. Louis, MO) according to manufacturer's protocol, treated with Turbo DNAse (Thermo Fisher Scientific, Waltham, MA) and then re-extracted with Tri-Reagent. RNA quality was assessed on Bioanalyzer Nanochip (Agilent, Santa Clara, CA) and only samples with RNA Integrity Number (RIN) ≥8.0 were used. Each sample was prepared from dozens of specimens (adult polyps and their tissues) or from hundreds of specimens (all younger developmental stages) in order to normalize for any individual variation. Those samples were analyzed on the nCounter platform (NanoString Technologies, Seattle, WA, USA; performed by Agentek Ltd., Israel) in technical triplicates, each made from a different batch of specimens following a previously described protocol (*Geiss et al., 2008*). In brief, for each transcript to be tested, two probes were generated and hybridized to the respective mRNA. The mRNAs were immobilized on a cartridge and the barcodes on one of the probes were counted by an automated fluorescent microscope. For normalization we used a geometric mean of the expression levels of 5 reference genes with stable expression across development. The genes were selected as follows: we calculated the Shannon entropy (as described in [*Schug et al., 2005*]) for each of 23,041 *Nematostella* genes based on normalized transcript abundance estimates for six time-points of *Nematostella* development (*Helm et al., 2013*). We then ranked the genes by entropy, which indicates minimal temporal change

in abundance, and from the top 20 chose five genes (NCBI Reference Sequences XM_001629766.1, XM_001628650.1, XM_001625670.1, XM_001640487.1 and XM_001624235.1) with complete gene models and mean abundance levels spanning the expected experimental range. Probe sequences, entropy scores and all raw and normalized nCounter read data are available in *Supplementary file 1*.

### Semi-quantitative MS/MS analysis

Hundreds of unfertilized eggs, 4 days old planulae, 9 days old primary polyps, adult males (five individuals), and adult females (five individuals) were lysed in 8M urea, 400 mM ammonium bicarbonate solution and centrifuged (22000 × *g*, 20 min, 4°C). Protein concentrations were measured with BCA Protein Assay Kit (Thermo Fisher Scientific). Ten μg of protein were reduced with DTT and alkylated with iodoacetamide. Tryptic digestion (0.3 μg trypsin/sample) was performed overnight at 37°C, followed by addition of 0.05% ProteaseMAX Surfactant (Promega Corp., USA) and further incubation for 1 hr at 37°C. The tryptic peptides were desalted on self-made C18 StageTips (*Rappsilber et al., 2007*). A total of 1.25 μg of peptides from each sample were injected into the mass spectrometer.

MS analysis was performed in four technical replicates using a Q Exactive Plus mass spectrometer (Thermo Fisher Scientific) coupled on-line to a nanoflow UHPLC instrument (Ultimate 3000 Dionex, Thermo Fisher Scientific). Eluted peptides were separated over a 180 min gradient run at a flow rate of 0.2 μl/min on a reverse phase PepMap RSLC C18 column (50 cm ×75 μm, 2 μm, 100 Å, Thermo Fisher Scientific). The survey scans (380–2,000 m/z, target value 3E6 charges, maximum ion injection times 50 ms) were acquired and followed by higher energy collisional dissociation (HCD) based fragmentation (normalized collision energy 25). A resolution of 70,000 was used for survey scans and up to 15 dynamically chosen most abundant precursor ions were fragmented (isolation window 1.6 m/z). The MS/MS scans were acquired at a resolution of 17,500 (target value 5E4 charges, maximum ion injection times 57 ms). Dynamic exclusion was 60 s.

Mass spectra data were processed using the MaxQuant computational platform, version 1.5.3.12 (*Cox and Mann, 2008*). Peak lists were searched against translated coding sequences of gene models from *N. vectensis*. The search included cysteine carbamidomethylation as a fixed modification and oxidation of methionine and N-terminal acetylation as variable modifications. Peptides with minimum of seven amino-acid length were considered and the required FDR was set to 1% at the peptide and protein level. Protein identification required at least two unique or razor peptides per protein group. The dependent-peptide and match-between-runs options were used. Relative protein quantification was performed using iBAQ values. MS/MS raw files as well results of MaxQuant analysis were deposited to the ProteomeXchange Consortium via the PRIDE (*Vizcaíno et al., 2016*) partner repository with the data identifier PXD008218.

### In situ hybridization (ISH)

Single and double ISH were performed as previously described (*Genikhovich and Technau, 2009a*; *Moran et al., 2013*). dFISH was performed also according to published protocols (*Nakanishi et al., 2012*; *Wolenski et al., 2013*) with tyramide conjugated to Dylight 488 and Dylight 594 fluorescent dyes (Thermo Fisher Scientific). In ISH and FISH, embryos older than 4 days were treated with 2 u/μl T1 RNAse (Thermo Fisher Scientific) after probe washing in order to reduce background. Stained embryos and larvae were visualized with an Eclipse Ni-U microscope equipped with a DS-Ri2 camera and an Elements BR software (Nikon, Tokyo, Japan). For each gene at least 20 specimens from each developmental stage were tested.

### Transgenesis

To generate transgenic constructs, we replaced the mCherry gene with mOrange2 (*Shaner et al., 2004*) and replaced the promoter sequence of the pNvT-MHC::mCH plasmid (*Renfer et al., 2010*). For the *NEP3* gene, we inserted to the plasmid 920 bp upstream of the transcription start site as well as the non-coding first exon, first intron and the part of the second exon that encodes the signal peptide of NEP3 (scaffold_7:1,219,288–1,221,320 of the *Nematostella* genome). For the NvePTX1 gene, we inserted to the plasmid 1033 bp upstream of the transcription start site as well as the non-coding first exon, first intron and the region of the second exon that encodes the signal peptide (scaffold_14:1,246,079–1,247,853 of the *Nematostella* genome). The constructs were injected with

the yeast meganuclease *I-SceI* (New England Biolabs, Ipswich, MA) to facilitate genomic integration (*Renfer et al., 2010*). Transgenic animals were visualized under an SMZ18 stereomicroscope equipped with a DS-Qi2 camera (Nikon).

## Immunostaining

Immunostaining was performed according to a previously described protocol (*Moran et al., 2012b*), employing a commercially-available rabbit polyclonal antibody against mCherry (Abcam) diluted to 1:400 and DAPI (Thermo Fisher Scientific) diluted to 1:500.

## Phylogenetics

Sequences of NEP and NvePTx1 protein families were retrieved using BLAST searches (*Altschul et al., 1990*) against NCBI's non-redundant nucleotide sequence database and the EdwardsiellaBase (*Stefanik et al., 2014*). Maximum-likelihood analysis was employed for the reconstruction of the molecular evolutionary histories. Trees were generated using PhyML 3.0 (*Guindon et al., 2010*), and node support was evaluated with 1000 bootstrapping replicates.

## Tentacle dissociation

Tentacles of *Nematostella* were dissociated using a combination of papain (2 mg/ml; Sigma-Aldrich: P4762), collagenase (2 mg/ml; Sigma-Aldrich: C9407) and pronase (4 mg/ml; Sigma-Aldrich: P5147) in DTT (1.3 mM) and PBS solution (1.8 mM $NaH_2PO_4$, 8.41 mM $Na_2HPO_4$, 175 mM NaCl, pH 7.4). The tentacles were incubated with the protease mixture at 22°C overnight. The tissues were then dissociated into single cells by flicking the tubes gently and then by centrifugation at 400 × $g$ for 15 min at 4°C, followed by resuspension in PBS.

## Purification of NEP3 from nematocysts

Lyophilized nematocysts were obtained from Monterey Bay Labs (Caesarea, Israel). 2.5 g of the nematocysts were discharged by incubation with 80 ml of 1% sodium triphosphate (Sigma-Aldrich). Following centrifugation (21,000 × g, 20 min), the crude extract was concentrated with Amicon centrifugal filters with 3 kDa cut off (Merck Millipore, Billerica, MA) to 2 ml volume, filtered through Amicon centrifugal filters with 50 kDa cut off (Merck Millipore) and used for further purification. At the first step, the extract was fractionated by size exclusion FPLC on a calibrated Superdex 75 column (60 × 1.6 cm, GE Healthcare, Little Chalfont, UK) in PBS buffer. Protein fractions with molecular weight less than 17.6 kDa were pooled and the PBS buffer was exchanged to 20 mM ethanolamine pH nine using Amicone centrifugal filters, cut off 3 kDa. At the second step, the SEC fractions were separated by anion exchange FPLC using a HiTrapQ HP column (1 ml, GE Healthcare) and a NaCl concentration gradient (0–750 mM NaCl in 30 column volumes, 20 mM ethanolamine pH 9.0). Fractions were analyzed by western blot with anti-Nep3 antibodies and positive ones were pooled. At the last step, Nep3 fragment was purified by reverse phase FPLC on a Resource RPC column (3 ml, GE Healthcare) using acetonitrile concentration gradient (8–60% $CH_3CN$ in 25 column volumes, 0.1% trifluoracetic acid). Fractions corresponding to individual peaks were collected and analyzed by western blot with anti-NEP3 antibodies.

## Recombinant expression and purification of toxins

A synthetic DNA fragment encoding the full NEP3 polypeptide was purchased from GeneArt (Regensburg, Germany). The fragment corresponding to the first domain of NEP3 between the Lys-Arg cleavage sites was amplified by PCR, cloned and expressed as a $His_6$-thioredoxin fusion protein in Shuffle T7 *Escherichia coli* strain (New England Biolabs).

Nv1 and NvePTx1 synthetic DNA fragments were purchased from Integrated DNA Technologies (Coralville, IA) and cloned into a modified pET40 vector (fragment encoding DSBC signal peptide was erased from it by Protein Expression and Purification facility of the Hebrew University to allow cytoplasmic expression of DSBC). Nv1 and NvePTx1 were expressed in BL21(DE3) *E. coli* (Merck Millipore) strain as fusions with $His_6$-DSBC.

The polyhistidine tag of the fusion proteins was used for purification from the *E. coli* lysate by nickel affinity FPLC. Purified fusion proteins were cleaved into two fragments by Tobacco Etch Virus (TEV) protease (room temperature, overnight) at a TEV protease cleavage site upstream the toxin

fragments. The recombinant toxins were then purified by reverse phase FPLC on a Resource RPC column (GE Healthcare) using an acetonitrile concentration gradient in 0.1% trifluoroacetic acid.

## Western blot

Custom polyclonal antibodies specific to the first domain of NEP3 were purchased from GenScript (Piscataway, NJ). Synthetic peptide containing the amino acid positions 47–91 was used as an antigen for immunization of two rats. The antibodies were affinity purified on a column coated with the antigen. Proteins were separated by electrophoresis on 10–20% gradient Tris-tricine gels (Bio-Rad, Hercules, CA) and consequently transferred to 0.2 um PVDF membranes (Bio-Rad). Membranes were blocked by 5% skim milk in TBST buffer (50 mM Tris base, 150 mM NaCl, 0.1% Tween 20, pH 7.6) and incubated with anti-NEP3 antibodies (1 ug/ml) in 5% Bovine serum albumin (BSA) in TBST buffer (4°C, overnight). This was followed by incubation with goat anti-rat IgG antibodies conjugated with horseradish peroxidase (0.1 ug/ml; Jackson ImmunoResearch, West Grove, PA) in 5% skim milk in TBST (room temperature, overnight). ECL reagent (GE Healthcare) was used for visualization of the protein bands interacting with anti-NEP3 antibodies. Chemiluminescence was recorded with an Odyssey Fc imaging system (LI-COR Biosciences, Lincoln, NE) and fluorescent size marker (Bio-Rad) was imaged on the same system.

## Animals for prey-predator and toxicity assays

*Danio rerio* larvae younger than 120 hr were generously provided by Dr. Adi Inbal (The Hebrew University Medical School). The usage of such young larvae does not require ethical permits according to the European and Israeli laws.

Fertilized *Fundulus* eggs from Kings Creek, VA (37°18 16.2"N 76°24 58.9"W) and Scorton Creek, MA (41°43′52.1"N 70°24′51.3"W) were kindly provided by Dr. Rafael Trevisan (Duke University) and Diane Nacci (Environmental Protection Agency), respectively. They were kept in 15 ‰ artificial sea water (ASW) at room temperature until hatching (around 2–3 weeks) and used immediately for behavioral analyses. Experiments on *F. heteroclitus* were performed under permit no. 17–018 granted by the Institutional Animal Care and Use Committee (IACUC) at the University of North Carolina at Charlotte according to ethical regulations of Office of Laboratory Animal Welfare (National Institutes of Health, USA).

The first batch of grass shrimps and adult *Fundulus* were collected at an estuary near Georgetown, SC (33°21′01.0"N 79°11′26.1"W). A second collection of grass shrimps were collected at Sippewissett Marsh, MA (41°35′22.9"N 70°38′17.0"W). Animals were transported to the lab and kept in 15 ‰ ASW until use. Interaction experiments were conducted in 15 ‰ ASW in 24-well plates or Pyrex bowls (~150 ml water, ~10 cm diameter), depending on size of animals and interactions being recorded. Grass shrimps were fed every day with either mussels or TetraMin tropical fish food (Tetra Holding, USA). *Fundulus* were fed twice daily with *Artemia* reared in the lab.

## Toxicity assays

To assess the toxicity of NEP3 and NvePTx1 on fish, 4 days old *D. rerio* larvae were incubated with 0.5 mg/ml peptides in 500 μl well. Each experiment was conducted in duplicates. 5 mg/ml BSA was used as a negative control. Three replicates were performed per treatment and each replicate included 5–7 larvae. Effect was filmed and monitored under an SMZ18 (Nikon) stereomicroscope after 5 min, 15 min, 1 hr, and 15–17 hr of incubation.

To assess the toxicity on grass shrimps, Nv1, NEP3, and NvePTx1 were dissolved in PBS buffer to 2.5–50 ng/μl concentration and 1 μl was injected into the abdomen from the ventral side for every 200 mg of shrimp mass. Ten shrimps were injected at each concentration.

## Interaction assays between different *Nematostella* life stages and potential prey and predator species

For testing the interaction of *Fundulus* with *Nematostella*, fish larvae were put in 24 well plates and preliminary screenings involving duplicate experimental observations were conducted for 5–10 isolated eggs, planula larva, and primary polyps of *Nematostella*. Based on these initial observations, we conducted additional observational experiments in triplicates, with each well containing five *Fundulus* larvae interacting with several egg packages or dozens of planula larvae. For each assay, the

fish were observed with portions of their interaction recorded to document how the fish responded to the egg packages or planula larvae. Additionally, for the isolated eggs, observational experiments were conducted with a single *Fundulus* larva in each well along with 10–15 eggs. The number of eggs were noted at the start of the experiment and observed every 2 hr over an 8 hr period. Observational experiments involving two *Fundulus* larvae and two adult *Nematostella* were conducted in duplicates using small Pyrex dishes (~50 ml of ASW) over a 2-day period, with and without substrate. For the adult *Fundulus*, observational experiments involving a single adult alongside eggs (>100), egg packages (4-5), larvae (>100), primary polyps (>100), and adult *Nematostella* polyps (3) were conducted over 48 hr in small Pyrex dishes (~200 ml water, ~10 cm diameter). Interactions were assayed in triplicates for shrimps, with a single shrimp interacting with different life stages: three egg packages, >100 eggs, >100 planula larvae, >100 primary polyps, and three adult *Nematostella* (with and without substrate). Across all observational experiments similar behavioral patterns were observed across experimental replicates, however, we were unable to identify exactly how many instances of an adverse reaction occurred during our observations. It was sometimes difficult to discern between the fish exhibiting sporadic swimming patterns, which we could not confidently link with them interacting with *Nematostella*.

For testing the potential effect of *Nematostella* primary polyps on *Fundulus* larvae, newly hatched *Fundulus* larvae (N = 10) were placed in a small glass aquarium (~3 ml) with 15 ‰ ASW. Fish behavior was recorded over 30 min per experiment using the Moticam 580 (Motic, Hong Kong, China), 15 min serving as a control and 15 min with a treatment. Freshly hatched *Artemia* were sacrificed by freezing and used as a positive control (N = 10). Silica beads 0.5 mm in diameter (BioSpec Products, Bartlesville, OK) were used as a negative control (N = 10). For each video, the recording was split into the following sections at these time scales: 0–5 min – acclimation, 5–15 min – control, 15–20 min – acclimation to treatment, 20–30 min – recorded behavior. The water container was split into three equal parts ('bottom', 'center' and 'top') based on the fish length. If the fish was in the 'center' with no portion of their body crossing either side time was not recorded as this was considered *no preference*. One-way ANOVAs were carried out separately for the upper and lower time points for each treatment. Specific treatments that were statistically significant were identified using the Tukey post hoc analysis. The results were plotted in GraphPad Prism version 7.00 for Windows (GraphPad Software, La Jolla, CA).

## Acknowledgements

We are grateful to Dr David Fredman (Department of Informatics, University of Bergen) for his invaluable help with quantitative and computational methods. We are also grateful to Dr Mario Lebendiker, Dr Tsafi Danieli and Yael Keren (Protein Expression and Purification Facilities of the Hebrew University) for their help with recombinant expression and chromatography and for the help of Dr Dana Reichmann and Dr Bill Breuer (Department of Biological Chemistry of the Hebrew University) for their help with mass spectrometry. We also thank Amy Klock and Matthew Kustra (Department of Biological Sciences, University of North Carolina, Charlotte) for their assistance with shrimp injections for the organismal assays. KS was supported by a Marie Skłodowska-Curie Individual Fellowship (654294). This work was supported by Israel Science Foundation grant no. 691/14 to YM, NSF award 1536530 to AMR, and Binational Science Foundation grant no. 2013119 to YM and AMR.

## Additional information

### Funding

| Funder | Grant reference number | Author |
| --- | --- | --- |
| Israel Science Foundation | Grant no. 691/14 | Yehu Moran |
| H2020 Marie Skłodowska-Curie Actions | Marie Skłodowska-Curie Individual Fellowship 654294 | Kartik Sunagar Yehu Moran |
| National Science Foundation | Award 1536530 | Adam M Reitzel |

| United States - Israel Binational Science Foundation | Grant no. 2013119 | Adam M Reitzel Yehu Moran |

The funders had no role in study design, data collection and interpretation, or the decision to submit the work for publication.

## Author contributions

Yaara Y Columbus-Shenkar, Formal analysis, Investigation, Visualization, Writing—original draft, Writing—review and editing; Maria Y Sachkova, Data curation, Formal analysis, Funding acquisition, Investigation, Methodology, Writing—original draft, Writing—review and editing; Jason Macrander, Formal analysis, Investigation, Methodology, Writing—original draft, Writing—review and editing; Arie Fridrich, Formal analysis, Investigation, Visualization, Methodology, Writing—review and editing; Vengamanaidu Modepalli, Investigation, Writing—review and editing; Adam M Reitzel, Resources, Supervision, Funding acquisition, Methodology, Writing—original draft; Kartik Sunagar, Conceptualization, Formal analysis, Funding acquisition, Investigation, Writing—review and editing; Yehu Moran, Conceptualization, Resources, Formal analysis, Supervision, Funding acquisition, Methodology, Writing—original draft, Project administration, Writing—review and editing

## Author ORCIDs

Yehu Moran [iD] http://orcid.org/0000-0001-9928-9294

## Ethics

Animal experimentation: Experiments on Fundulus heteroclitus were performed under permit no. 17-018 granted by the Institutional Animal Care and Use Committee (IACUC) at the University of North Carolina at Charlotte according to ethical regulations of Office of Laboratory Animal Welfare (National Institutes of Health, USA).

## Decision letter and Author response

Decision letter https://doi.org/10.7554/eLife.35014.025
Author response https://doi.org/10.7554/eLife.35014.026

# Additional files

## Supplementary files

• Supplementary file 1. Information of nCounter probe sequences, nCounter raw and normalized data and entropy list for *Nematostella* transcripts. This file is related to *Figure 1E-G*.
DOI: https://doi.org/10.7554/eLife.35014.017

• Supplementary file 2. Results of MaxQuant analysis of tandem mass spectrometry on *Nematostella* lysates from different developmental stages.
DOI: https://doi.org/10.7554/eLife.35014.018

• Transparent reporting form
DOI: https://doi.org/10.7554/eLife.35014.019

## Major datasets

The following dataset was generated:

| Author(s) | Year | Dataset title | Dataset URL | Database, license, and accessibility information |
|---|---|---|---|---|
| Maria Y Sachkova, Yehu Moran | 2018 | Dynamics of venom composition across a complex life cycle | http://proteomecentral. proteomexchange.org/ cgi/GetDataset?ID= PXD008218 | Publicly available at ProteomeXchange (accession no. PXD00 8218) |

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
