## [Decision Letter]

Thank you for submitting your work entitled "Dynamics of venom composition across a complex life cycle" for consideration by *eLife*. Your article has been reviewed by two peer reviewers, and the evaluation has been overseen by a Reviewing Editor and a Senior Editor. The reviewers have opted to remain anonymous.

Our decision has been reached after consultation between the reviewers. Based on these discussions and the individual reviews below, we regret to inform you that your work will not be considered further for publication in *eLife*.

Although this manuscript reports on the interesting finding that toxin composition of the starlet sea anemone *Nematostella vectensis* is highly dynamic, there are several major concerns that limit the potential significance of this work. These are amply presented in the reviews below.

Reviewer #2:

In this manuscript, Columbus-Shenkar et al. report an interesting finding: the toxin composition of the starlet sea anemone *Nematostella vectensis* is highly dynamic, with a potential potassium channel blocker, NvePTx1, being exclusively expressed during larval stages and different NEP3 family toxins being expressed in different subpopulations of mature nematocysts. The authors applied experimental approaches including proteomics, transcriptomics, toxicology and gene profiling to support their findings. However, several major concerns limit the potential significance of this work.

1) As the authors point out, similar studies focusing on venom composition dynamics have already been reported in both invertebrates (cone snail)[1] and vertebrates (rattlesnakes)[2]. And like the majority of phylum Mollusca, cone snails also possess a complex life cycle, with a planktonic veliger stage and a highly predatory adult stage. Similar to previous studies, this manuscript provides another descriptive analysis of venom composition changes during different life stages, but fails to touch on the biological relevance or the molecular mechanism behind this phenomenon.

2) The toxicology experiments were meant to demonstrate the toxicity of *Nematostella* proteins NvePTx1 and NEP3, which was a main argument made by the authors. However, both experiments were carried out and analyzed in a crude manner and definitely require improvement. Synthesis of biologically active toxic peptides has been proven difficult, as the formation of disulfide bridges between evolutionarily conserved cysteine residues is rather inefficient in *E. coli* culture system (even with Shuffle and BL21 strains, in which the cytosolic disulfide bond formation is enhanced)[3, 4]. More importantly, the existence of multiple cysteine residues leads to potential disulfide bond formation at different positions. The authors purified both toxins directly from bacterial culture without further analysis of peptide structure, and tried to demonstrate the toxicity of each peptide by incubating them at a single concentration with zebrafish larvae. A series of questions need to be answered: i) What proportion or isoform of purified peptides are biologically active? And why do these peptides only function at such a high concentration compared to previous studies [4, 5]?

ii) Is zebrafish a suitable subject to test toxins from *Nematostella* (whose diet likely consists of invertebrates such as small crustaceans and mollusks)?

iii) Is it appropriate to conclude on the neurotoxicity of the peptides simply based on abnormal tail contraction phenotype of zebrafish larvae? These experiments lead to more confusion, rather than providing functional evidence to support the argument that both genes truly function as toxins.

3) The quality of data in general needs to be improved. Images from double in situ hybridization experiments (Figure 4) were meant to demonstrate partial co-localization of different toxins and nematocyst marker gene Ncol3. But the poor quality of these images fails to provide enough cellular resolution to support these claims. Also, the FISH pattern of the same gene NEP3 at primary polyp stage varies from panel to panel, and is not comparable to the double in situ pattern shown in Figure 5. The same problem exists with the Ncol3 pattern, which was used as a reference marker, in Figure 4, primary polyp stage, row1 versus row 4.

References:

1. Safavi-Hemami H, Siero WA, Kuang Z, Williamson NA, Karas JA, Page LR, et al. Embryonic toxin expression in the cone snail Conus victoriae: primed to kill or divergent function? The Journal of biological chemistry. 2011;286(25):22546-57.

2. Gibbs HL, Sanz L, Chiucchi JE, Farrell TM, Calvete JJ. Proteomic analysis of ontogenetic and diet-related changes in venom composition of juvenile and adult Dusky Pigmy rattlesnakes (Sistrurus miliarius barbouri). J Proteomics. 2011;74(10):2169-79.

3. Rosano GL, Ceccarelli EA. Recombinant protein expression in *Escherichia coli*: advances and challenges. Frontiers in microbiology. 2014;5:172.

4. Luo S, Zhangsun D, Harvey PJ, Kaas Q, Wu Y, Zhu X, et al. Cloning, synthesis, and characterization of alphaO-conotoxin GeXIVA, a potent alpha9alpha10 nicotinic acetylcholine receptor antagonist. Proceedings of the National Academy of Sciences of the United States of America. 2015;112(30):E4026-35.

5. Warmke JW, Reenan RA, Wang P, Qian S, Arena JP, Wang J, et al. Functional expression of *Drosophila* para sodium channels. Modulation by the membrane protein TipE and toxin pharmacology. J Gen Physiol. 1997;110(2):119-33.

Reviewer #3:

Summary

Venom research has primarily focused on pharmacological analyses of venoms and toxins from adult stages of species. While ontogenetic variation has been reported in the venoms of some species of snakes and cone snails, the influence of development and ecology on venom production is poorly understood, particularly in organisms with complex life cycles. This study uses an integrative approach (behavior, transcriptomics, biochemistry, transgenesis) to study venom production across all life stages in the starlet sea anemone *Nematostella vectensis.* Specifically, the authors quantify the spatial and temporal expression of known and novel toxins across the life stages of *Nematostella*.

Significance

The findings of this study are important for improving our understanding of developmental processes, the evolution of venom and its role mediating interspecific interactions, and the impact of venom-mediated interactions on biodiversity.

Isolated toxins from venomous animals (cone snails, scorpions, sea anemones, spiders) have been used for years to study ion channels. Less attention has been paid to the influence of development and ecology on venom production and composition. Most examples are from snake or cone snail species. Few studies integrate development, physiology and ecology of venoms and toxins, mainly because there are few organisms that can be studied both in natural habitats and in the lab, especially organisms with complex life cycles. Distinct patterns of toxin gene expression in *Nematostella* that track developmental stages raise intriguing questions and open new avenues for further exploration of toxins and toxin-producing genes. For example, this study lays the foundation for future work aimed at understanding the selective forces that shape venom production and delivery, the link between functional differences in toxin activity and fitness, and how these processes change during development.

Moreover, this research has the potential to uncover novel peptides or molecules that could be used in the discovery of novel drugs and drug targets. There are many channels for which selective blockers or modulators have yet to be identified. For example, there are approximately 90 genes that encode K^+^ channels. Identification of novel toxins from *Nematostella* suggests this venom may be a rich source of small molecules and peptides that can be used to selectively manipulate K^+^ channels for drug discovery.

Comments/Suggestions

Overall, this is a really nice study that uses multiple techniques to quantify toxin expression across life stages and different adult tissues. The different assays (gene expression, biological activity of toxins, ISH/dFISH, etc.) are rigorous, and the authors do a good job of considering alternative hypotheses. The authors do not need to do more experiments. However, there are some areas of the manuscript that could be improved.

Suggestions for the Discussion

This study provides robust data that show differences in toxin-gene and toxin expression across developmental stages and adult tissues. This suggests that venom composition changes across developmental stages. While differences in the expression of toxins across developmental stages and tissues suggest there may be functional differences in venom that provide optimal activity at a particular life stage, in a particular ecological setting, or in response to different interspecific interactions, the authors should be careful of over interpreting their data to suggest functional differences for which there is little support. It is plausible that differential toxin expression and venom composition are the result of differential selective forces across developmental stages. However, this study did not explicitly show that differences in venom composition cause different functional effects on planula larvae versus adult polyps. Moreover, while venoms that differ in composition may provide optimal activity at different life stages or in different tissues, the authors should be careful to avoid statements that suggest evolution is purposeful and results in adaptations that are structurally and functionally optimal. For example: 1) "[…] each arsenal of toxin might be tailored for different biological needs."; 2) "Hence, there is a clear advantage in using a highly adapted venom in each developmental stage."

Regarding the use of venom for defense versus prey capture – this is an intriguing area of venom physiology and evolution. Localized expression of toxin genes or toxins suggests functional specificity. However, the authors should avoid "attributing" functional specificity to localized expression. It is also not clear from the data shown in this study that NvePTx1 is solely a defensive toxin. NvePTx1 was identified as a homolog of the type 5 K^+^ channel blocker BCsTx3 from sea anemone based on sequence similarity. To confirm toxic activity, recombinant NvePTx1 was incubated with zebra fish larvae, which died by the end of the 16 hour incubation period, compared to control larvae incubated with no toxin. The ISH staining showed a developmental pattern consistent with eggs expressing NvePTx1, and expression patterns changing as the eggs developed. But unless I missed this in the data or reference in another publication, has it been shown that eggs expressing incubated with predators actually kills the predators? Regarding the NvLysin1b pore-former toxin, while expression in the pharynx and mesenteries suggests it is for prey capture, perhaps it also functions to help break food down to aid digestion.

Perhaps the authors could revise their discussion to distinguish between conclusions that can be drawn from the data, and speculation about differential venom function and future directions.

It would be helpful to include a little more information about the toxins and their biological activity. References are provided, but this requires that the reader search for information that could be provided in a concise manner. For example, the authors could briefly define "pore-forming toxin". Is this a toxin that lyses cells versus a toxin that blocks ion channel pores or manipulates gating mechanisms? Toxin experts won't be bothered by this but other readers will appreciate brief definitions.

[Editors’ note: the author responses to the first round of peer review follow.]

---

## [Author Response]

Although this manuscript reports on the interesting finding that toxin composition of the starlet sea anemone Nematostella vectensis is highly dynamic, there are several major concerns that limit the potential significance of this work. These are amply presented in the reviews below.

In order to address the idea that the different venom compositions produced during distinct developmental stages of *Nematostella* is facilitating different interactions with other species, we conducted lab experiments where two predatory species that come from native *Nematostella* habitat, the killifish *Fundulus heteroclitus* and the grass shrimp *Palaemonetes* sp., were exposed to different developmental stages of *Nematostella*. Those interactions were tested multiple times with nearly perfect reproducibility. All these details, including the exact description of the interactions and number of animals and replicates are now provided in the Results section and Materials and methods. All the interaction results are provided in the new table, Table 1. To provide the reader with visual description of some notable forms of interaction we also provide videos. These videos describe typical interactions that were recorded multiple times (n>3) in independent experiments. The results of the interaction of killifish larvae with primary polyps of *Nematostella*, originally caught on video, are provided in the new Figure (Figure 6) and are quantified and statistically significant. The combination of all of these results show that indeed antagonistic interactions of *Nematostella* change with development. We do not claim to fully solve the very complex picture of *Nematostella* ecology in the field, but we believe that these new results provide a strong support to the idea that the vast changes of venom composition enable different interspecific interactions. We would also like to note that this hypothesis is also supported by the great differences we reveal by transcriptomics, proteomics, and multiple cytology assays. It is likely that the appearance and disappearance of whole cell populations that produce different toxins across development documented here at such magnitude for the first time, is not a neutral process and requires significant metabolic resources. We suggest that these noticeable cellular and biochemical transformations were evolutionarily shaped by the different interspecific interactions. Those interactions obviously also depend on the vastly different sizes of the developmental stages and their motility or lack thereof and we do not claim that venom is the only factor shaping them. However, we do suggest that our results support the idea that the costly changes in venom composition are most probably correlated with predation ecology and were selected via interspecific interactions.

Reviewer #2:[…] 1) As the authors point out, similar studies focusing on venom composition dynamics have already been reported in both invertebrates (cone snail)[1] and vertebrates (rattlesnakes)[2]. And like the majority of phylum Mollusca, cone snails also possess a complex life cycle, with a planktonic veliger stage and a highly predatory adult stage. Similar to previous studies, this manuscript provides another descriptive analysis of venom composition changes during different life stages, but fails to touch on the biological relevance or the molecular mechanism behind this phenomenon.

This is in our opinion the single most critical point in the letter, and also the most subjective one as novelty is a matter of opinion. We will do our best to explain in what points our paper is different and novel compared to the previous works where differential developmental expression of toxins was indicated:

1) First of all, it is important to consider what previous works have established: the work from 2011 of Gibbs et al. [1] reported only subtle changes in venom composition between young and adult rattlesnakes, as the authors themselves state (quote: “Juvenile snakes fed from birth with mice, lizards, or frogs showed little evidence for an ontogenetic shift in venom composition from 5 to 26 months in terms of substantial changes in the relative abundance of major classes of venom toxins. However, there were fine-scale changes in the relative abundance of D49-PLA₂ 15, PI-SVMPs, and PIIISVMP 28, and a decline in the abundance of other PIII-SVMPs.”). The work by Safavi-Hemami on cone snails [2], showed that venom changes between embryos and adults of the species *Conus victoriae*. However, this work showed only two developmental stages, used only two adults as the source of venom and used non-quantitative (normal reverse transcription PCR) at the transcriptomic level.

2) In contrast to the previous works, the transcriptomic part of our work was done in triplicates on 10 developmental stages and multiple tissues and used thousands of embryos and dozens of adults that normalize any individual variation. This is important as individual variability might have contributed to the differences documented in the previous works (see reviews [3, 4]). Further, our *Nematostella vectensis* lab population is highly inbred, controlling for genetic variation in a way that the experiments on cone snails and snakes or any other study on field-collected animals could never control. This is an important point as individuals of various venomous species were shown to exhibit much variation in venom expression due to both genetic and environmental conditions. We now mention this point in the Introduction section.

3) Further, our work is far more quantitative than the two previous works, as Nanostring nCounter technology was shown in multiple studies to be arguably the most accurate technique for transcript level measurements as it is non-enzymatic. We have now also added a semi-quantitative tandem mass-spectrometry analysis to support our transcriptomic findings also at the proteomic level.

4) Most importantly, our work also shows highly complex tissue and cellular dynamic localization that to our knowledge was never shown before by others.

5) We do not agree with the reviewer on the point that our findings fail to touch on biological significance, as we show in the manuscript for the first time that sea anemone planulae larvae are capable of rapidly paralyze and kill other animals and that they discharge their nematocysts in this process. Coupled with the highly different venom composition between young and adult stages, we do touch a new biological relevance of venom variation and venom utilization by larva. The cone snail work [2], as elegant as it is, did not do anything parallel to this point.

6) We now also support the biological relevance of our findings by presenting detailed analysis of interactions between different *Nematostella* developmental stages and two potential predators from the same habitat (see Table 1, Figure 6 and videos).

7) Lastly, this might be an obvious point, but our work is performed in a cnidarian, and not on cone snails or snakes that are separated from our model organism by 600 million years. In our opinion, saying that it just repeats previous findings in other systems (which is wrong for all of the above reasons), is analogical to saying that finding a phenomenon in mouse is not exciting because it was already described in *Drosophila*.

2) The toxicology experiments were meant to demonstrate the toxicity of Nematostella proteins NvePTx1 and NEP3, which was a main argument made by the authors. However, both experiments were carried out and analyzed in a crude manner and definitely require improvement. Synthesis of biologically active toxic peptides has been proven difficult, as the formation of disulfide bridges between evolutionarily conserved cysteine residues is rather inefficient in E. coli culture system (even with Shuffle and BL21 strains, in which the cytosolic disulfide bond formation is enhanced)[3, 4]. More importantly, the existence of multiple cysteine residues leads to potential disulfide bond formation at different positions. The authors purified both toxins directly from bacterial culture without further analysis of peptide structure, and tried to demonstrate the toxicity of each peptide by incubating them at a single concentration with zebrafish larvae. A series of questions need to be answered:i) What proportion or isoform of purified peptides are biologically active? And why do these peptides only function at such a high concentration compared to previous studies [4, 5]?ii) Is zebrafish a suitable subject to test toxins from Nematostella (whose diet likely consists of invertebrates such as small crustaceans and mollusks)?iii) Is it appropriate to conclude on the neurotoxicity of the peptides simply based on abnormal tail contraction phenotype of zebrafish larvae? These experiments lead to more confusion, rather than providing functional evidence to support the argument that both genes truly function as toxins.

As we described in the Materials and methods section, the two toxins were purified by metal ion chromatography, and more importantly by reverse phase chromatography. The fish larvae were incubated only with highly purified toxins. We provide in Author response image 1 figure showing the reverse phase results of purified toxins and it is easy to see that the vast majority of toxin molecules represent a single isoform as demonstrated by the highly symmetric peak in the chromatogram. From our substantial experience with recombinant cysteine-rich peptides (please see numerous relevant publications by Drs. Moran and Sachkova where this technique was successfully applied), this is a clear evidence for a high degree of homogeneity. The pure peptides were also assayed on electrospray mass spectrometry, yielding masses that fit the formation of disulfide bonds. Solving the 3D structures of these peptides in X-ray or NMR is completely out of the scope of the current manuscript and will be futile unless compared to structures of the native peptides. However, the structure of a Type V sea anemone toxins was never described and they have no homology to other toxin families, whereas NEP3 is only very-far related to the ShK or BgK toxins, making any structural comparisons highly hypothetical. Such an experimental work will take years and is again completely unrelated to the core of the current manuscript and its focus. Further, even in the work by Luo et al. [5] that the reviewer is providing as an example, the three synthetic isoforms of the cone snail toxin all show quite potent activity on the nicotinic acetylcholine receptor tested (alpha9alpha10, please see Figure 2 in that paper).

**Author response image 1. respfig1:** Chromatograms of the recombinant sea anemone toxins used in the zebrafish experiments. The purified toxins were re-run on a Resource RPC 3 ml reverse phase column (GE Healthcare).

The reason for the high concentration that is needed for killing the zebrafish might be that penetration is not very efficient and takes a long time as the fish larvae are incubated with the toxins. Alternatively, some toxins are just not very active alone but are highly synergic with other venom components (see for example [6]). Whatever the reason is, the toxins were able to kill the fish in a few hours whereas the control group, incubated with a non-toxic protein, was alive and well even after overnight incubation. We have now expanded our zebrafish experiments by using more specimens and monitoring more frequently the toxic effects.

Is zebrafish the right model for our question? Well, are lab mice good model for scorpion, conesnail and spider toxins? That’s a highly debatable topic, with many possible answers. In our paper that focuses on completely different topics, we believe that documenting toxicity is enough for supporting our claim that these are toxins. It is important to remember that this proof is corroborated by the fact that the peptides, show homology to previously characterized sea anemone neurotoxins and were found to be expressed in venom-related nematocytes and gland cells. In our opinion, zebrafish is a suitable model. After all, it represents teleost fish and vertebrates in general because even if it is a freshwater fish from southeastern Asia, it shares the vast majority of ion channel subtypes with other fish species. Please also note that *Nematostella* lives in brackish lagoons that host fish larvae and young-feeding developmental stages of fish. Further, fish can serve as predators and not necessarily as prey, which is anyway much more relevant in the case of NvePTx1 (expressed only in non-feeding stages) and NEP3 (expressed in both feeding and non-feeding stages). We also provide now in our new results (Table 1 and videos) evidence that killifish larvae from the same habitat try to feed on *Nematostella* eggs and planulae, but fail to consume them and release them unharmed.

The reviewer is providing the works of Warmke et al. [7] and Luo et al. [5] as examples for high toxin activity. In the first work the authors applied a sea anemone sodium channel modulator (ATX-II) on a fruit fly sodium channel expressed in frog oocytes and in the second work the authors applied a cone snail acetylcholine receptor antagonist on DRG neurons from rat and on mammalian acetylcholine receptors heterologously expressed in frog eggs. Of course one cannot compare apples and oranges, but if the reviewer accepts these highly artificial systems as biologically relevant, our zebrafish experiment is not so bad after all. Further, very high activity of toxins is not a strong proof for ecological relevance, as very strong toxic effects on ecologically irrelevant species (such as sea anemones and flies or cone snails and rats) and off-target effects of toxins, yet with high activity, on pharmacologically irrelevant targets are both well documented in the field of toxinology (see review [3]).

For all the above reasons, we believe that our toxicity tests on fish are relevant and provide sufficient proof for the activity of NEP3 and NvePTx1 as toxins. However, we agree with the reviewer that the claim for neurotoxic effects based only on tail twitching should be toned down and this is why we use the word “might”.

3) The quality of data in general needs to be improved. Images from double in situ hybridization experiments (Figure 4) were meant to demonstrate partial co-localization of different toxins and nematocyst marker gene Ncol3. But the poor quality of these images fails to provide enough cellular resolution to support these claims. Also, the FISH pattern of the same gene NEP3 at primary polyp stage varies from panel to panel, and is not comparable to the double in situ pattern shown in Figure 5. The same problem exists with the Ncol3 pattern, which was used as a reference marker, in Figure 4, primary polyp stage, row1 versus row 4.

Honestly, we are struggling to understand the basis for this remark. We suspect that this has to do with quality of PDF transferred to the reviewer as there is a very significant decrease in quality between our high resolution TIF pictures and the PDF created by the *eLife* system. We provide in Author response image 2 an example for the reviewer so they can clearly see a cellular-resolution figure demonstrating the partially-overlapping expression patterns of *NEP3* and *NEP3-like*. This figure is part of our original Figure 4 that we uploaded to the *eLife* system. Additionally, we further solve this problem by adding to Figure 4 in the manuscript zoomed in panels to better show the co-localization. Further, we also suggest to take the following important point into consideration: In situ hybridization patterns of nematocyte markers or toxins will never be identical between panels as there is considerable animal to animal and even layer to layer variation like with any single cell pattern, i.e., a salt-and-pepper pattern, in *Nematostella*. In most animals (basically almost any animal species which is not a nematode) even under full genetic homogeneity, embryos are not 100% identical to one another in their spatiotemporal expression patterns. For example, if we would use in situ hybridization to localize a gene in a fly brain that is expressed in only a subpopulation of the neurons, every specimen would be slightly different from the other. That’s perfectly normal.

**Author response image 2. respfig2:** Double Fluorescent in Situ Hybridization experiment demonstrating that the expression patterns of NEP3 and NEP3-like only partially overlap (yellow signal demonstrates overlapping expression of the two genes).

Reviewer #3:[…] This study provides robust data that show differences in toxin-gene and toxin expression across developmental stages and adult tissues. This suggests that venom composition changes across developmental stages. While differences in the expression of toxins across developmental stages and tissues suggest there may be functional differences in venom that provide optimal activity at a particular life stage, in a particular ecological setting, or in response to different interspecific interactions, the authors should be careful of over interpreting their data to suggest functional differences for which there is little support.

We agree with the reviewer and hence toned down our interpretations regarding function.

It is plausible that differential toxin expression and venom composition are the result of differential selective forces across developmental stages. However, this study did not explicitly show that differences in venom composition cause different functional effects on planula larvae versus adult polyps. Moreover, while venoms that differ in composition may provide optimal activity at different life stages or in different tissues, the authors should be careful to avoid statements that suggest evolution is purposeful and results in adaptations that are structurally and functionally optimal. For example: 1) "[…] each arsenal of toxin might be tailored for different biological needs."; 2) "Hence, there is a clear advantage in using a highly adapted venom in each developmental stage."

Of course, we completely agree with the reviewer that evolution is not purposeful and we did not mean to convey such an erroneous idea. We rephrased the text to avoid such misunderstandings.

Regarding the use of venom for defense versus prey capture – this is an intriguing area of venom physiology and evolution. Localized expression of toxin genes or toxins suggests functional specificity. However, the authors should avoid "attributing" functional specificity to localized expression. It is also not clear from the data shown in this study that NvePTx1 is solely a defensive toxin. NvePTx1 was identified as a homolog of the type 5 K^+^ channel blocker BCsTx3 from sea anemone based on sequence similarity. To confirm toxic activity, recombinant NvePTx1 was incubated with zebra fish larvae, which died by the end of the 16 hour incubation period, compared to control larvae incubated with no toxin. The ISH staining showed a developmental pattern consistent with eggs expressing NvePTx1, and expression patterns changing as the eggs developed. But unless I missed this in the data or reference in another publication, has it been shown that eggs expressing incubated with predators actually kills the predators?

This is the only point where we somewhat disagree with reviewer #3. Because NvePTx1 is found only in the egg, embryos that do not feed, and in the gonads of adult females (but not in adult males), we think we can safely assume it is not used for predation, but solely for defense. However, we agree that this is still a suggestion as we did not prove directly that NvePTx1 is used in defense and hence we edited the text accordingly.

Regarding the NvLysin1b pore-former toxin, while expression in the pharynx and mesenteries suggests it is for prey capture, perhaps it also functions to help break food down to aid digestion.

We completely agree and we included this idea in a previous publication (18) and also now in the revised text.

Perhaps the authors could revise their discussion to distinguish between conclusions that can be drawn from the data, and speculation about differential venom function and future directions.

We think that this is a good idea and this is why we declare hypothetical parts as such (now highlighted in blue for the referee).

It would be helpful to include a little more information about the toxins and their biological activity. References are provided, but this requires that the reader search for information that could be provided in a concise manner. For example, the authors could briefly define "pore-forming toxin". Is this a toxin that lyses cells versus a toxin that blocks ion channel pores or manipulates gating mechanisms? Toxin experts won't be bothered by this but other readers will appreciate brief definitions.

Thank you for bringing up this idea. That’s an excellent way to make the paper more accessible to readers from other fields.

1. Gibbs HL, Sanz L, Chiucchi JE, Farrell TM, Calvete JJ (2011) Proteomic analysis of ontogenetic and diet-related changes in venom composition of juvenile and adult Dusky Pigmy rattlesnakes (Sistrurus miliarius barbouri). *J Proteomics* 74: 2169-79

2. Safavi-Hemami H, Siero WA, Kuang Z, Williamson NA, Karas JA, Page LR, MacMillan D, Callaghan B, Kompella SN, Adams DJ*, et al.* (2011) Embryonic toxin expression in the cone snail Conus victoriae: primed to kill or divergent function? *J Biol Chem* 286: 22546-57

3. Casewell NR, Wüster W, Vonk FJ, Harrison RA, Fry BG (2013) Complex cocktails: the evolutionary novelty of venoms. *Trends Ecol Evol* 28: 219-229

4. Sunagar K, Morgenstern D, Reitzel AM, Moran Y (2016) Ecological venomics: How genomics, transcriptomics and proteomics can shed new light on the ecology and evolution of venom. *J Proteomics* 135: 62-72

5. Luo S, Zhangsun D, Harvey PJ, Kaas Q, Wu Y, Zhu X, Hu Y, Li X, Tsetlin VI, Christensen S*, et al.* (2015) Cloning, synthesis, and characterization of alphaO-conotoxin GeXIVA, a potent alpha9alpha10 nicotinic acetylcholine receptor antagonist. *Proc Natl Acad Sci U S A* 112: E4026-35

6. Cohen L, Lipstein N, Gordon D (2006) Allosteric interactions between scorpion toxin receptor sites on voltage-gated Na channels imply a novel role for weakly active components in arthropod venom. *FASEB J* 20: 1933-5

7. Warmke JW, Reenan Ra, Wang P, Qian S, Arena JP, Wang J, Wunderler D, Liu K, Kaczorowski GJ, Van der Ploeg LH*, et al.* (1997) Functional expression of *Drosophila* para sodium channels. Modulation by the membrane protein TipE and toxin pharmacology. *J Gen Physiol* 110: 119-33

8. Moran Y, Praher D, Schlesinger A, Ayalon A, Tal Y, Technau U (2013) Analysis of soluble protein contents from the nematocysts of a model sea anemone sheds light on venom evolution. *Mar Biotechnol (NY)* 15: 329-39

9. Lapidot M, Pilpel Y (2006) Genome-wide natural antisense transcription: coupling its regulation to its different regulatory mechanisms. *EMBO Rep* 7: 1216-22

10. Yelin R, Dahary D, Sorek R, Levanon EY, Goldstein O, Shoshan A, Diber A, Biton S, Tamir Y, Khosravi R*, et al.* (2003) Widespread occurrence of antisense transcription in the human genome. *Nat Biotechnol* 21: 379-86

11. Ho MR, Tsai KW, Lin WC (2012) A unified framework of overlapping genes: towards the origination and endogenic regulation. *Genomics* 100: 231-9

12. Bradshaw B, Thompson K, Frank U (2015) Distinct mechanisms underlie oral vs aboral regeneration in the cnidarian Hydractinia echinata. *ELife* 4: e05506

13. Kraus Y, Aman A, Technau U, Genikhovich G (2016) Pre-bilaterian origin of the blastoporal axial organizer. *Nat Comm* 7: 11694

14. Sinigaglia C, Busengdal H, Leclère L, Technau U, Rentzsch F (2013) The Bilaterian Head Patterning Gene six3/6 Controls Aboral Domain Development in a Cnidarian. *PLoS Biol* 11: e1001488

15. Watanabe H, Schmidt HA, Kuhn A, Hoger SK, Kocagoz Y, Laumann-Lipp N, Ozbek S, Holstein TW (2014) Nodal signalling determines biradial asymmetry in Hydra. *Nature* 515: 1125

16. Levitan S, Sher N, Brekhman V, Ziv T, Lubzens E, Lotan T (2015) The making of an embryo in a basal metazoan: Proteomic analysis in the sea anemone *Nematostella vectensis. Proteomics* 15: 4096-104

17. Lotan T, Chalifa-Caspi V, Ziv T, Brekhman V, Gordon MM, Admon A, Lubzens E (2014) Evolutionary conservation of the mature oocyte proteome. *EuPA Open Proteomics* 3: 2736

18. Moran Y, Fredman D, Szczesny P, Grynberg M, Technau, U (2012) Recurrent horizontal transfer of bacterial toxin genes to eukaryotes. *Mol Biol Evol* 9: 2223-30